

# Estimating observation and model error variances using multiple data sets

Richard A. Anthes[1] and Therese Rieckh[1,2]

[1]COSMIC Program Office, University Corporation for Atmospheric Research, Colorado, U.S.A.

[2]Wegener Center for Climate and Global Change, University of Graz, Graz, Austria

*Correspondence to:* Richard Anthes (anthes@ucar.edu)

**Abstract.** In this paper we show how multiple data sets, including observations and models, can be combined using the "*N*-
cornered hat method" to estimate vertical profiles of the errors of each system. Using data from 2007, we estimate the error
variances of radio occultation, radiosondes, ERA-Interim and GFS model data sets at four radiosonde locations in the tropics
and subtropics. A key assumption is the neglect of error covariances among the different data sets, and we examine the
consequences of this assumption on the resulting error estimates.

## 1 Introduction

Estimating the error characteristics of any observational system or model is important for many reasons. Not only are these
errors of scientific interest, they are important for data assimilation systems and numerical weather prediction. In many modern
data assimilation schemes, observations of a given type are weighted proportionally to the inverse of their error variance (e.g.
Desroziers and Ivanov, 2001).

Kuo et al. (2004) and Chen et al. (2011) used the difference between radio occultation (RO) observations of a variable X (e.g.
refractivity) and short-range model forecasts of X to estimate the error of the RO observations, using the concept of *apparent*
or *perceived errors*, defined by

$$X_{AE} = X_{RO} - X_{fcst} \qquad\qquad (1)$$

where $X_{AE}$ is the apparent error of the RO observation of X and $X_{RO}$ and $X_{fcst}$ are the RO observations and model forecast
values of X respectively.

The error variance $\sigma_a^2$ of the apparent error is given by





$$\sigma_a^2 = \sum X_{AE}^2 / N \qquad\qquad (2)$$

where N is the number of samples of observed and model RO at the same location and time.

The relationship between the apparent error variance $\sigma_a^2$, the observational error variance $\sigma_o^2$, and the forecast error variance $\sigma_f^2$ is given by

$$\sigma_a^2 = \sigma_o^2 + \sigma_f^2 - 2COV(X_{RO}X_{fcst}), \qquad\qquad (3)$$

where the COV term is the error covariance between the observations and the forecasts. If the error variance of the forecast $\sigma_f^2$ is estimated independently, the observational error variance can be estimated from the apparent error variance, under the assumption that the observational errors are uncorrelated with the forecast errors (in which case the COV term in Eq. (3) is zero).

$$\sigma_a^2 = \sigma_o^2 + \sigma_f^2 \qquad\qquad (4)$$

We note that the apparent errors are the same as the (O-B) (observation minus background) or innovations as used in data assimilation methods and studies (Chen et al., 2011).

As discussed by Kuo et al. (2004) and Chen et al. (2011), the forecast error variance can be estimated by two alternative methods, the "NMC method" (Parrish and Derber, 1992) or the Hollingsworth and Lönnberg (1986) method. Kuo et al. (2004) used both methods to estimate the observational errors of RO refractivity using the NCEP AVN model. Chen et al. (2011) used the NMC method and Weather and Research Forecast Model (WRF) to estimate the forecast error variance and then the RO refractivity error variance.

In this paper, we estimate the error variances of multiple data sets using the "*N*-cornered hat" method (Gray and Allan, 1974). Unlike the apparent error method, this method does not require independent estimates of the error variance of a forecast; it uses the differences between combinations of *N* data sets. The "*N*-cornered hat" method is described in Appendix A along with the related "triple collocation method" (Stoffelen, 1998). The data sets may be either different model or observational

data and estimates of the error variances of all the data sets are computed by the method. We compare three observational data sets (two versions of radio occultation retrievals and radiosondes) and two model data sets at four locations in the tropics and subtropics to estimate the error variances of all five data sets. We find that the results are consistent with each other and with previous error estimates, where available.





## 2 Discussion of data sets

We use five data sets from an entire year (2007) in this study. Rieckh et al. (2017, hereafter Paper 1), extensively studied the
properties of these data sets and their daily variability over 2007 in the tropical and sub-tropical western Pacific. They are
described in more detail there, but are summarized briefly here for convenience.

We chose 2007 for the year of our study because the number of COSMIC (Constellation Observing System for Meteorology,
Ionosphere and Climate) RO observations was near a maximum at this time. Because our primary interest in Paper 1 was the
evaluation of water vapor observations and model analyses in challenging tropical and subtropical environments, we chose
one radiosonde (RS) station in the deep tropics and three Japanese stations in the subtropics. Because of our focus on water
vapor, we carry out the analysis from 1000 to 200 hPa.

### 2.1 ERA-Interim

The ERA-Interim (hereafter ERA) reanalysis is global model reanalysis produced by the European Centre for Medium-Range
Weather Forecasts (ECMWF) (Dee et al., 2011). Information about the current status of ERA-Interim production, availability
of data online, and near-real-time updates of various climate indicators derived from ERA-Interim data, can be found at
https://www.ecmwf.int/en/research/climate-reanalysis/reanalysis-datasets/era-interim

The ERA assimilates both RS and RO data for the entire year of 2007; hence some correlation of model, RS, and RO errors is
likely. However, there are many other observations going into the ERA reanalysis and so the model correlations with any one
observational data set is likely to be small.

### 2.2 NCEP Global Forecast System (GFS)

The Global Forecast System (GFS) is a forecast model produced by the National Centers for Environmental Prediction
(NCEP). Data are available for download through the NOAA National Operational Model Archive and Distribution System
(NOMADS). Forecast products and more information on GFS are available at
https://www.ncdc.noaa.gov/data-access/model-data/model-datasets/global-forcast-system-gfs .
Prior to January 2003, the GFS was known as the Aviation model (AVN), which was one of the models used by Kuo et al.
(2004) in their estimation of RO errors using the apparent error method.
The GFS assimilated RS observations for the entire year 2007, but began assimilating RO data on 1 May, 2007, along with
many other changes to the model and analysis system (Cucurull and Derber, 2008; Kleist et al., 2009). Thus the GFS and RS
and RO errors are also likely correlated to some degree. However, we computed vertical profiles of the correlation coefficients





for RO and GFS refractivity, temperature, specific humidity and relative humidity in the two months before and after May 1, 2007 when the GFS started assimilating RO data and found little differences, so the error correlations between RO and GFS are likely small.

## 2.3 Radio occultation observations

The RO observations used in this study are obtained from the UCAR COSMIC Data Analysis and Archive Center (CDAAC). Two methods for estimating the temperature and water vapor from the RO refractivity are used. In the *direct method*, the GFS temperature is used in the Smith and Weintraub (1953) equation

$N = 77.6\ P/T + 3.73 \times 10^5 e/T^2$                           (5)

to compute water vapor pressure e from the observed N and GFS temperature T.

A one-dimensional variational (1D-VAR) method is also used to estimate temperature and water vapor pressure from N. The 1D-VAR method uses an a-priori state of the atmosphere (background profile) and an observed RO refractivity to minimize a

quadratic cost function. At CDAAC, an ERA-Interim profile is used as background, which is interpolated to the time and location of the RO observation (accounting for tangent point drift during the occultation). The humidity retrieval allows an error for both *T* and *e*, but only a very small error for bending angle/refractivity. Specific humidity q is then computed from the derived water vapor pressure e.

**2.4 Radiosonde observations**

Radiosonde data from Guam and three Japanese stations are used in this comparison. The radiosondes are given on nine main pressure levels between 1000 hPa and 200 hPa, plus additional levels if atmospheric conditions are variable. The four stations use the following sensors: Guam: VIZ/Sippican B2; Ishigakijima: Meisei; Minamidaitojima: Vaisala RS92; and Naze: Meisei.

They are launched twice daily in the hour before noon and midnight, UTC.

Guam is located in the deep tropics at 13.7°N 144.8°E. Ishigahijima (hereafter called Ishi), Minamidaitojima (hereafter called Mina), and Naze are located relatively close together in the western Pacific subtropics south of Japan and northeast of Taiwan:
Naze: Naze/Funchatoge (Kagoshima) 28.4°N 129.4°W

Mina: Minami-daitojima (Okinawa) 25.6°N 131.5°W
Ishi: Ishigakijima (Okinawa) 24.2°N 124.5°W




The RO observations used in the comparisons are those within 600 km and 3 hours of the radiosonde launch. The model observations closest to the four radiosonde stations respectively are used. Corrections for the spatial separation are made using the model data. The details are described by Gilpin et al. (2017, in preparation).

5    The refractivity for the radiosonde and model data is computed from Eq. (5) using the pressure, temperature and water vapor from these data. Normalized differences are computed for all combinations of the data sets (RO-ERA, RO-GFS, GFS-ERA, RS-ERA, RS-GFS, RS-RO). The ERA annual mean for 2007 at each RS station is used to normalize the differences in the data sets associated with that station. We consider the differences among the five data sets for four variables: refractivity (N), temperature (T), specific humidity (q) and relative humidity (RH).

**2.5 Number of samples**

The number of samples is limited by the number of RO observations that are within the co-location criteria of three hours and 600 km. Figure 1 shows the number of RO profiles that meet these criteria during 2007 at Mina (the numbers at Ishi and Naze

15   are similar) and Guam. The number of samples at the Japanese stations is a maximum of approximately 900 at 300 hPa. It decreases rapidly above 300 hPa to zero at 200 hPa because of the decrease in the number of RS observations above 300 hPa. The number decreases to about 100 at 950 hPa at the three Japanese stations. At Guam, the number ranges from a maximum of about 500 at 200 hPa to about 50 at 950 hPa at Guam. Thus the effect of the limited sample size will be greatest for the Japanese stations above 300 hPa and for all four stations below 900 hPa where the sample size is less than 500.

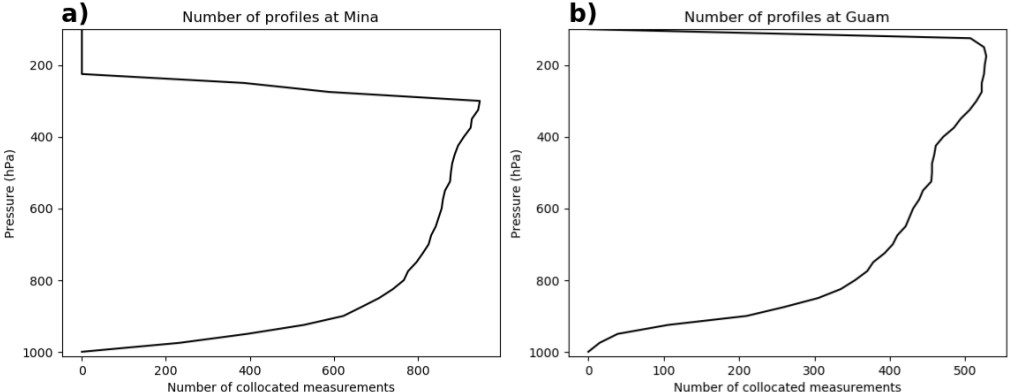

**Figure 1: Number of RO refractivity profiles matching the co-location criteria for Mina (a) and Guam (b) during the year 2007. These are also the number of samples in the calculations of the estimated error variances for the five data sets (RO Direct and RO 1D-VAR, RS, ERA and GFS).**





**2.6 Mean ERA profiles for 2007 and example of profiles and normalized difference profiles**

Before showing the statistical comparisons of the normalized differences between the data sets and their estimated errors, we present the mean ERA profiles of q, RH, T and N at Mina and Guam for the year 2007 (Figure 2). The standard deviations are

5    shown by the shading around each mean profile. As shown by Figure 2, the water vapor (especially relative humidity) shows the greatest variability over the year. The variability in specific humidity, temperature and refractivity is greater at Mina, which is located in the subtropics, than Guam, which is located in the deep tropics.

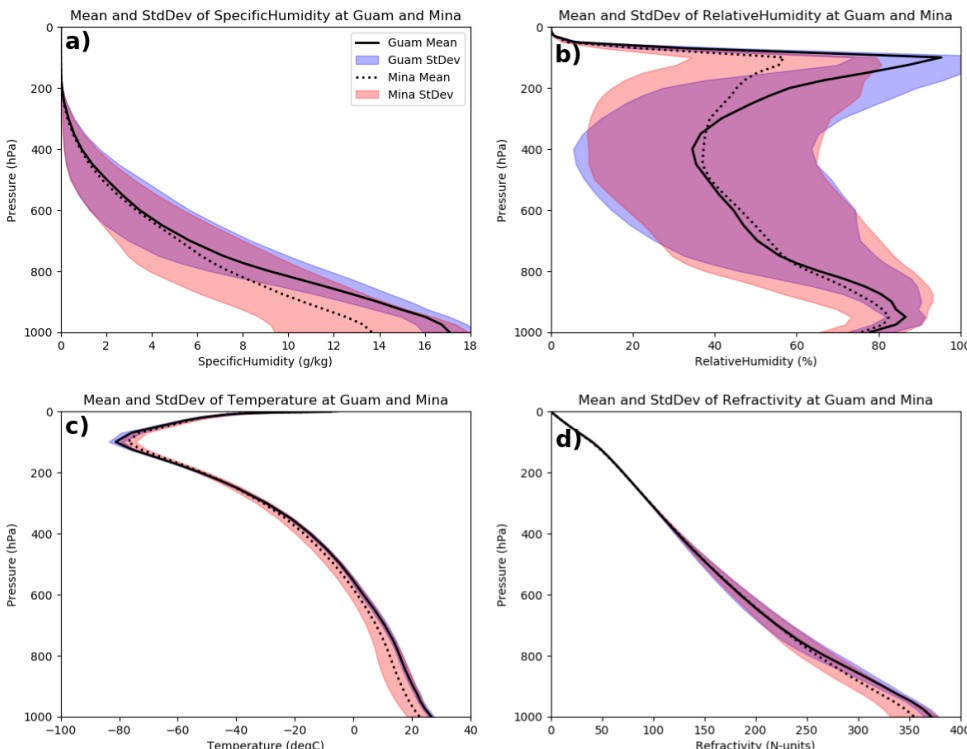

**Figure 2: The mean ERA profiles over 2007 at Guam and Mina of specific humidity q (a), relative humidity RH (b), temperature T**

10   **(c), and refractivity N (d). The standard deviations about the mean profiles are indicated by the shading.**

We next present a single example of soundings from the five data sets, to illustrate how the profiles of the normalized differences of the variables (which we use in all the following calculations) compare to the actual profiles. Figure 3 illustrates the q, RH, T and N profiles from 13 January 2007 at approximately 00 GMT and Figure 4 illustrates the corresponding profiles

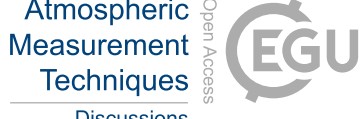



of the normalized differences of the variables from ERA, for example $(q - q_{ERA})/\text{CLIMO}$, where CLIMO is the 2007 mean ERA value of the variable.

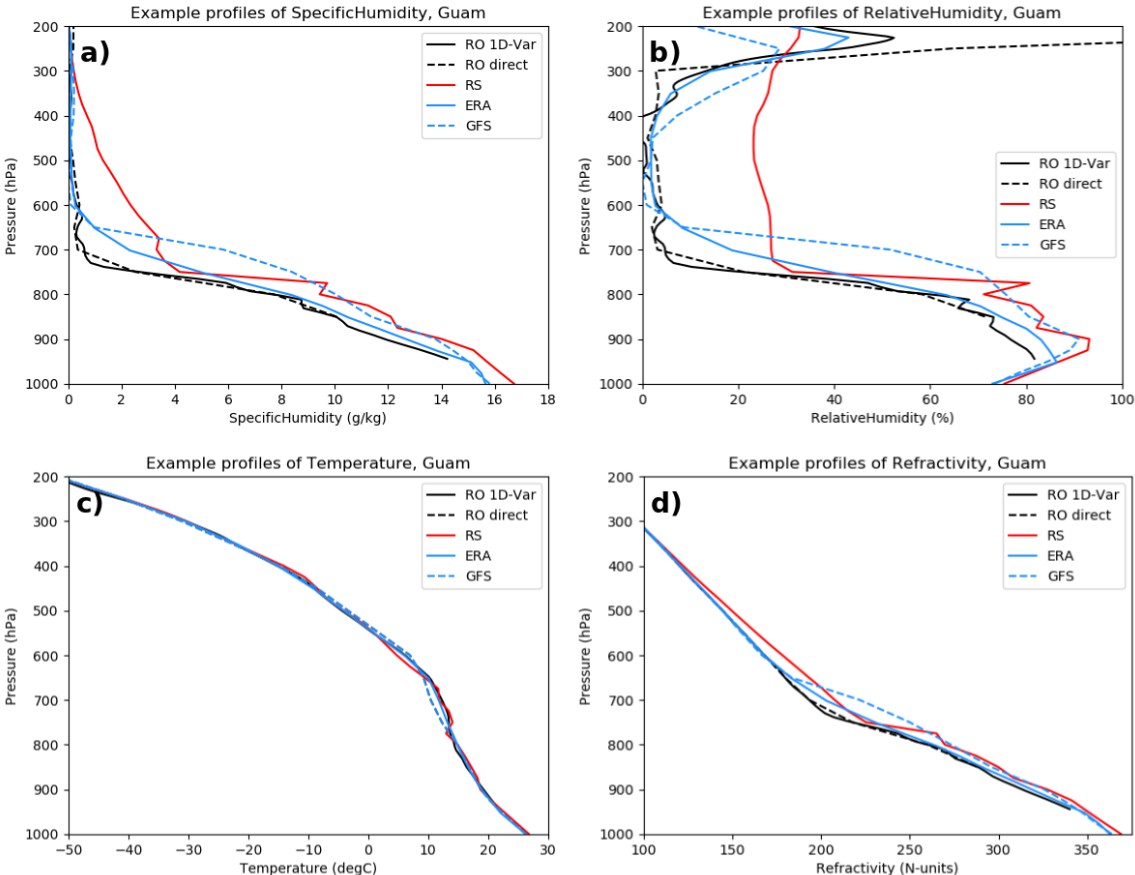

5  **Figure 3: Profiles of specific humidity (a), relative humidity (b), temperature (c) and refractivity (d) for the five data sets for 13 January, 2007 at 00:23 UTC.**




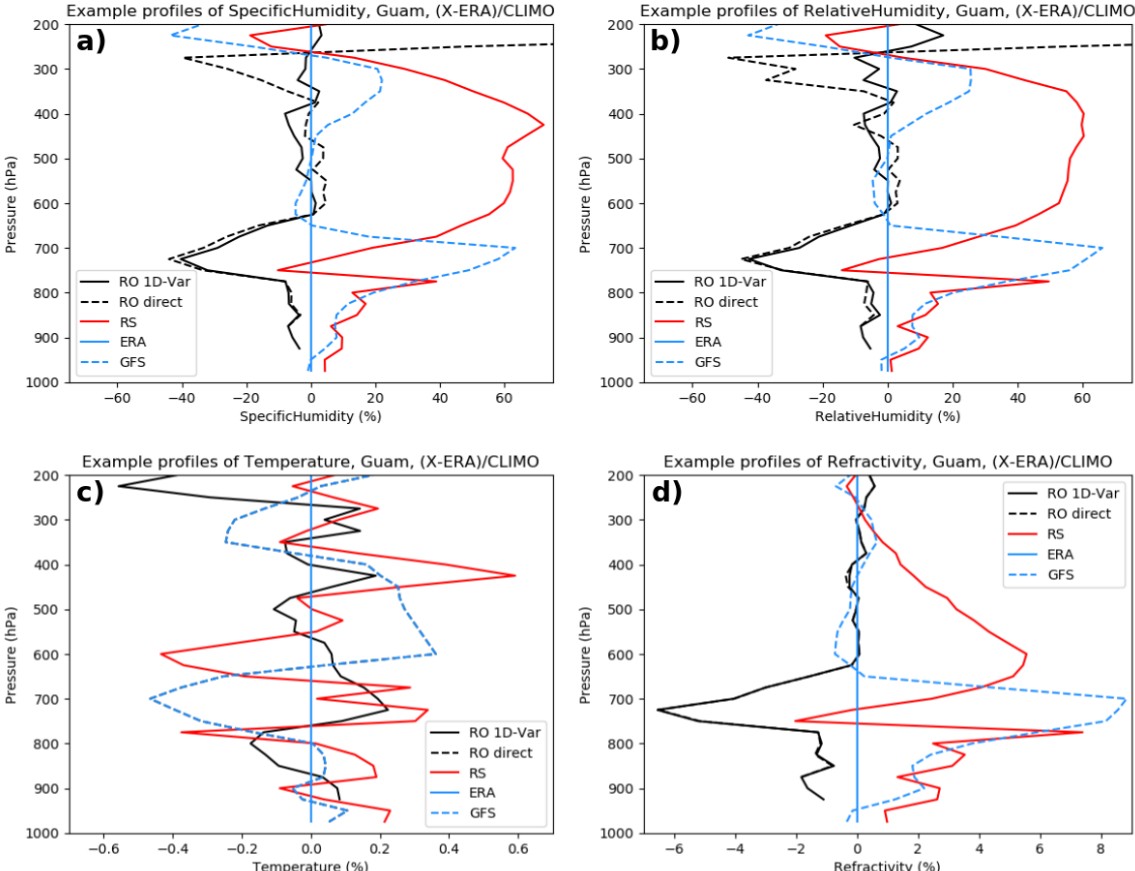

**Figure 4: Same as Figure 3 except for normalized differences from ERA.**

A comparison of Figures 3 and 4 shows that the normalized difference profiles highlight the similarities and differences of the

5   five data sets better than the actual profiles, especially in the upper troposphere. The magnitudes of the normalized differences

are the same order of magnitude at all levels, whereas the differences in the actual profiles can vary by more than an order of

magnitude from the lower to the upper troposphere. Figure 4 shows that typical percentage differences between data sets are

~50% for q and RH, 0.5% for T, and 5% for N.

**3 Derivation of error variances**

10   In this section we summarize the derivation of the equations relating the error variances and covariances among the five data

sets. Details, references, and a discussion of the limitations is given in Appendix A.





The error variance of a variable X (e.g. q, RH, T or N) is defined as

$$VAR(X) = [\sum(X-X_t)^2]/N \tag{6}$$

where $X_t$ is the true (but unknown) value of X and the summation is over N samples.

As shown in Appendix A, we can derive three different linearly independent equations for estimating the error variance of any data set, assuming that the error covariances among all the data sets are negligible compared to the differences in the observed

10   mean square (MS) differences between the data sets. Three more equations can be derived by using linear combinations of the first three. Appendix A provides examples of the error estimates using the six and three equations. For example, the three complete (and exact) linearly independent solutions for estimating the error variance of RO are

$$2\ VAR(RO) = MS(RO\text{-}ERA) + MS(RO\text{-}GFS) - MS(GFS\text{-}ERA)$$
$$+2\{COV(RO,ERA)+COV(RO,GFS)-COV(GFS,ERA)\} \tag{7}$$

$$2\ VAR(RO) = MS(RO\text{-}ERA) + MS(RO\text{-}RS) - MS(RS\text{-}ERA)$$
$$+2\{COV(RO,ERA)+COV(RO,RS) - COV(RS,ERA)\} \tag{8}$$

$$2\ VAR(RO) = MS(RO\text{-}GFS) + MS(RO\text{-}RS) - MS(RS\text{-}GFS)$$
$$+2\{COV(RO,GFS)+COV(RO,RS) - COV(RS,GFS)\} \tag{9}$$

where RO (or ERA, GFS, RS) corresponds to the value of X as estimated by RO (or ERA, GFS, RS) and MS denotes the mean square difference between the values from two data sets (e.g. RO-ERA).

We use Eqs. (7)-(9) to provide three independent estimates of VAR(RO) by neglecting the COV terms in each equation. The assumption that the error covariances are small compared to the difference in variances between the data sets is similar to the assumption used in the apparent error method that the errors of the observations and model forecasts are uncorrelated. Of course in general the COV terms are not zero; thus we will examine the validity of this assumption by checking whether the

30   various estimates of the error variances from the three equations are consistent with each other and reasonable compared to other independent studies that estimate error variances in other ways.

The same procedure can be used to derive three equations for estimating the error variances for the other three data sets, RS, ERA, and GFS (equations not shown here).

So for each of the five data sets, RO Direct and 1D-VAR, RS, ERA, and GFS, there are three independent ways to estimate their respective error variances. We note that it is possible that the estimated error variances from any of the three equations



are negative because of the neglect of the COV terms and the small sample size, especially above 300 hPa for the Japanese stations and below 800 hPa for all four stations (Figure 1).

**4 Comparison with previous studies for RO refractivity**

We first compute the estimated error variance for RO refractivity using GFS and ERA data for comparison with the Kuo et al. (2004) and Chen et al. (2011) estimates of RO error variance to illustrate the three cornered hat method. In an analogy to the apparent error Eq. (4), with RO being the observation and ERA being the forecast

MS(RO-ERA) = VAR(RO) + VAR(ERA)                    (10)

which is Eq. (1) in Appendix A with neglect of the COV terms.

We compute MS(RO-ERA) from the RO and ERA data sets (analogous to the apparent error variance $\sigma_a^2$ in Eq. (4)) and plot

its square root as the black line in Figure 5. Then we estimate VAR(RO) using Eq. (7) and the data sets (RO-GFS) and (GFS-ERA), along with the apparent error MS(RO-ERA), neglecting the COV terms.

The square root of VAR(RO) gives the blue curve labelled STD(RO-True) in Figure 5. Finally, the ERA error variance (analogous to the forecast error) is obtained by subtracting VAR(RO) from MS(RO-ERA) using Eq. (10) above (pink line in

Figure 5).





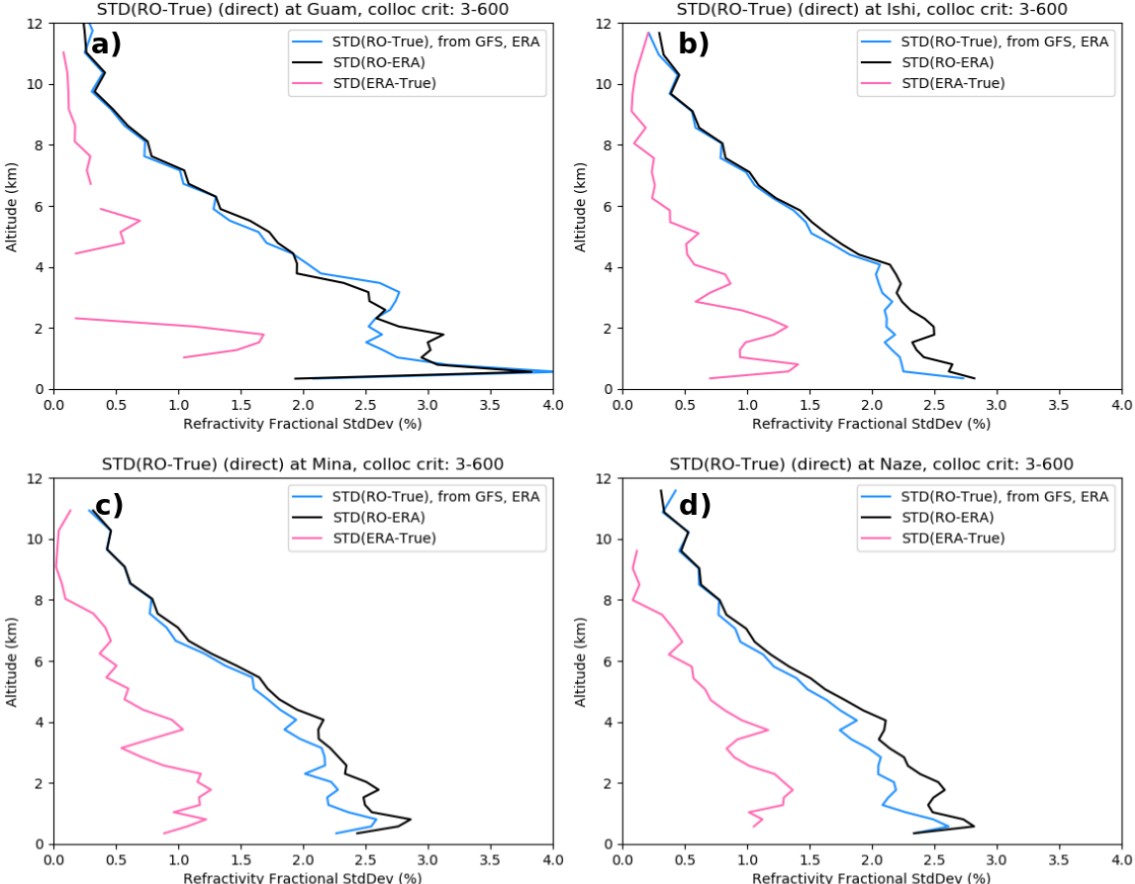

**Figure 5: Standard deviations of the apparent error STD(RO-ERA) (black line), estimated RO error STD(RO-True) computed from Eq. (7) (blue line) and ERA error STD(ERA-True) (pink line) for refractivity at a) Guam, b) Ishi, c) Mina, and d) Naze.**

The results shown in Figure 5 are quite similar to those from Kuo et al. (2004, Figure 13) and Chen et al. (2011, Figure 3d) who used different models and different data sets. The STD of normalized RO refractivity errors are a maximum of between 2.0 and 2.5% near the surface, decreasing to about 0.5% at 10 km. The shape of the profiles between 0 and 2 km is also similar in the two methods, with a small local minimum in the profile at about 1 km. These similarities give credibility to both methods.



**5 Calculation of the error variance terms using the multiple data sets**

This section shows the estimated error variances for N, q, T and RH at one of the four stations (Mina) for the five data sets and summarizes the results for the other three stations (Naze, Ishi and Guam).

**5.1 Results for Mina**

The following plots show the estimated error variances computed from Eq. (7), (8), or (9). Two RO data sets (Direct and 1D-VAR) are considered one at a time using the other three data sets. Thus we have two sets of error estimates for each data set: one using the Direct RO with RS, ERA and GFS, and one using the 1D-VAR RO with RS, ERA and GFS. In the following
plots, results for six error estimates are indicated by the color. Darker colors correspond to the three results using the 1D-VAR RO, lighter colors correspond to the three results using the Direct RO.

Figure 6 shows the results for specific humidity. Error variances are shown rather than STD because they are easier to interpret
using the three equations used to derive them and because the STD are undefined for the occasional negative estimated error variance. Figure 6a shows the q error variance profiles for the two RO data sets (Direct and 1D-VAR). The direct method [use of GFS temperature in Eq. (5)] shows a steady increase of error variance with height, from about 100 %$^2$ (STD ~10%) at 950 hPa to 800 (STD ~28%) at 500 hPa and 2000 (STD ~45%) at 300 hPa. This is expected since the refractivity contains little information on water vapor above about 400 hPa and we are using an independent estimate of temperature, with no constraints
on the water vapor retrieval. The q error variance profile for RO using the 1D-VAR method is similar to that of the direct method below 500 hPa, but reaches a maximum at about 500 hPa of about 500%$^2$ (STD ~22%) and then decreases toward zero at 200 hPa. The 1D-VAR method uses the ERA-Interim fields as background and thus constrains the water vapor profile retrieval at high altitudes. It is notable that the three equations used to estimate the error variance profiles agree closely and the difference among the three estimates is much smaller than the differences in the mean profiles using the two RO retrieval
methods.

The specific humidity error variance profiles associated with the radiosonde at Mina (Figure 6b) show a similar behavior as the direct RO, with a steady increase with height, exceeding a VAR of 2000%$^2$ (STD of ~45%) at 400 hPa. The STD of the RS are slightly larger than the two RO estimates below 600 hPa. The error variance estimates using the Direct RO (orange)
and 1D-VAR RO (red) are similar.

The error variance profiles from the two model sets (Figure 6c,d) are quite different. The GFS error variance is less that the RO Direct and radiosonde at all levels, and also less than the RO 1D-VAR except above 300 hPa. Although there is more scatter, especially in the upper troposphere, the ERA profiles are different from all the other data sets in that they show only a




small increase of error variance with height, from a variance near zero at the surface to up to a mean of about 100%²
(STD~10%) at 200 hPa. The ERA profiles contain examples of the estimated error variances becoming negative. This is
because the true values of the error variances are close to zero and so neglect of the error covariance terms can produce small
negative values.

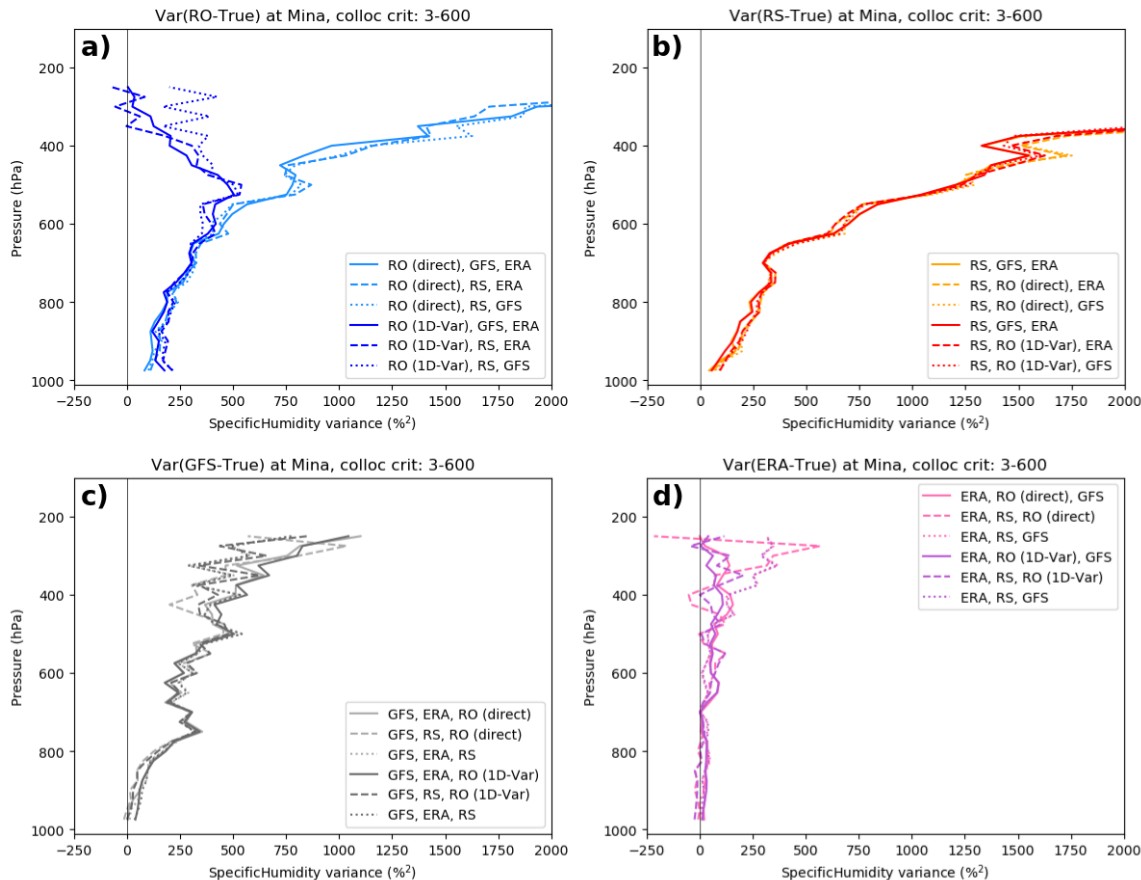

Figure 6: Estimated error variances (% squared) of specific humidity at Mina: a) RO, b) RS, c) GFS and d) ERA.



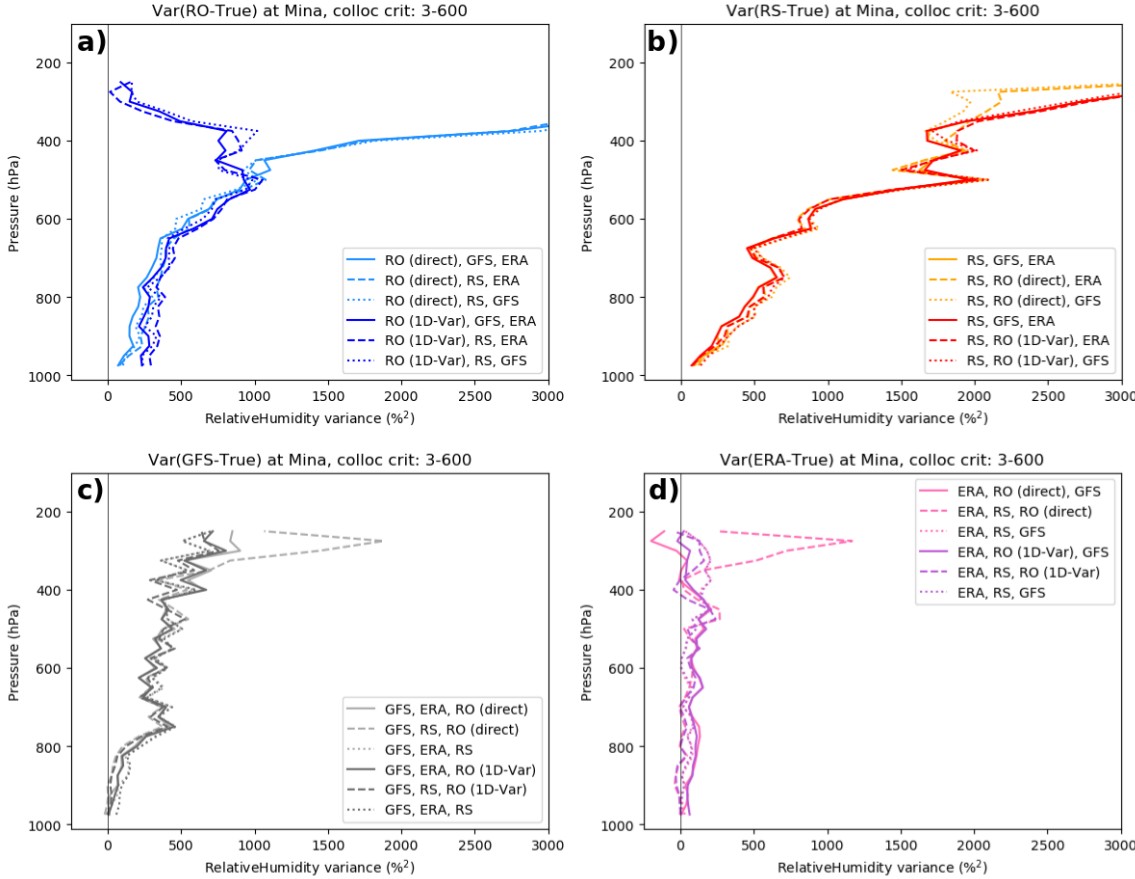

**Figure 7: Estimated error variances (% squared) of relative humidity at Mina: (a) RO, (b) RS, (c) GFS and (d) ERA.**

Figure 7 shows the estimated error variances of relative humidity. As with specific humidity, there is consistency among the
5   estimates for the different data sets. The general behavior of the RH error variance profiles is similar to those for q, as might
be expected because the percentage variability of water vapor is greater than that of temperature at this subtropical location.
Again, the estimated error variances of the RO derived RH are less than those of the radiosondes in the lower troposphere. The
GFS error variances are smaller than the RO and RS variances, except for the RO 1D-VAR profile above 300 hPa, which is
constrained by the ERA observations in the upper troposphere. The ERA error variances are significantly smaller than the
10  other data sets, averaging between 50 and 200 %$^2$ (STD 7-14%) throughout the troposphere.



Figure 8 shows the estimated error variances of temperature. Because the RO-Direct retrieval uses the exact GFS temperature, the results for the direct retrieval (light blue) using the (RO, GFS, ERA) and (RO, RS and GFS) are not meaningful in Figure 8a (they are identically zero). The result from Eq. (8) (RO, ERA and RS), given by the dashed light blue line in Figure 8a is valid, but in reality this is an estimate of the GFS T error variance, and it is in fact very similar to the profiles in Figure 8c.

The RO 1D-VAR results for temperature from all three equations give somewhat larger results (Figure 9a, dark blue profiles). The estimated error variance profiles oscillate between 0.1 and 0.3 %$^2$ (STD 0.3 to 0.55%). For a temperature of 300K, these correspond to 0.9 to 1.65 K.

The radiosonde temperature error variances (Figure 8b) vary between 0.05 and 0.15 %$^2$ (STD 0.2% to 0.4% or 0.6 to 1.2K for T=300K). The GFS temperature error variances are a little lower, averaging around 0.05-0.10 (STD 0.2-0.3%), while the ERA estimated temperature error variances average close to zero (Figure 8d).

Figure 9 shows the estimates of the normalized refractivity errors for the five data sets. There is more spread in the refractivity
estimates compared to those of the other variables, especially in the lower troposphere where the estimates vary between about 4 and 9 %$^2$ (STD 2-3%) for the two RO variances. Recall that the RO-Direct N are the observed RO N as provided by CDAAC while the RO 1D-VAR N are modified based on the background (ERA) N. The average of the N error variances for the radiosondes (Figure 9b) shows a maximum of ~10 %$^2$ (STD ~3.2%) around 900 hPa. The GFS error variance profiles show a maximum around 750 hPa of ~8 %$^2$ (STD ~2.8%). The ERA profiles show the smallest errors, with a maximum in the lower
troposphere of an average of ~2%$^2$ (STD~1.4%). All data sets show a decrease of error variance to less than 0.5%$^2$ (STD <0.7%) at 400 hPa. The reason for the large scatter in estimates of N below about 800 hPa may be related to errors in N caused by super-refraction in the lower troposphere, which occurs often in the tropics and subtropics. Super-refraction causes a negative N bias, which may lead to larger error covariances in this layer. The smaller number of RO samples below 800 hPa (Figure 1) may also be a factor.

Figure 10 shows the mean of the six estimates of the error variances of the five data sets for q, RH, T and N at Mina. The standard deviation about these means is shown by the shaded areas. These figures show clearly the significant differences among the error variance estimates of the five data sets. In Figure 10a, the error variance for specific humidity is greatest for the radiosonde (red and orange profiles) and least for the ERA profiles. As discussed earlier, the mean of the RO 1D-VAR
retrieval reaches a maximum at about 550 hPa and then decreases back toward zero as it becomes constrained by the background profile at high levels. Figures 10b-d show the mean profiles of error variance for relative humidity, temperature and refractivity. The relative humidity profiles are similar to the specific humidity profiles. The ERA errors are the smallest, followed by GFS, the RO and finally the radiosondes. The temperature error variance profiles show that the ERA errors are very close to zero throughout the entire troposphere. The GFS and RO profiles are fairly constant with height at values of



about 0.05 and 0.1%$^2$ respectively (0.22% or 0.7K at 300K and 0.32% or 0.9K at 300K respectively. The RS shows an oscillating error variance profile ranging between 0.1 and 0.3%$^2$ (0.3% and 0.5% or 0.9K and 1.5K at 300K respectively). Finally, the refractivity profiles show the greatest variability, but the mean profiles are still quite distinct. ERA again shows the lowest errors, followed by GFS, RO and RS.

It is difficult to find previous results for RS temperature and specific humidity error variances. However, previous studies comparing RO with RS and models indicate that our estimates are reasonable and consistent with these studies. Ho et al. (2017) found STD between RO and RS pairs for many RS types of about 1.5K in the layer 200-20 hPa, where RO temperatures are most accurate (Table 2 in Ho et al., 2017). This value corresponds to the apparent error between RS and RO, which is larger

than the RS error. The estimated RS temperature error variances from 200 to 100 hPa in Figure 10c is about 0.15%$^2$, which corresponds to a STD of 0.39% or 0.9K for a mean temperature of 230 K. Ladstädter et al. (2015) compared high-quality GRUAN RS to RO globally and for a tropical station (Nauru) and subtropical station (Tateno, Japan) from 2002 to 2013. They found temperature STD of about 0.5K for Nauru and 0.5-0.8K at Tateno averaged over the layer 800 hPa to 300 hPa. For specific humidity, they found STD between RO and RS of about 10% increasing to about 40% in the upper troposphere. In

our calculations for Guam and the three subtropical Japanese stations our estimates for STD of q are similar (Figure 10a and Appendix B, Figure B1), ranging from about 10% at 900 hPa to 45% at 300 hPa.

Ho et al. (2010) compared COSMIC RO observations to ECMWF analyses and several types of radiosondes for the period August-November, 2006. They found mean specific humidity STD of RO-ECMWF of ~0.5 g/kg and RO-RS (Meisei) of ~0.9

g/kg. From their plots of the vertical profiles of the STD, these numbers are typical for the layer 800-500 hPa, which, given the normalization values from the four RS stations in our study (Figure 3) of about 9 g/kg at 800 hPa and 2 g/kg at 500 hPa, correspond to % STD/VAR values of ~6%/36%$^2$ at 800 hPa and 25%/625%$^2$ at 500 hPa for ECMWF and ~10%/100%$^2$ at 800 hPa and 45%/2025%$^2$ at 500 hPa for Meisei RS. These values are similar to the estimates of the GFS analysis and RS analysis for the Japanese stations shown in Figure B1 of Appendix B.

Vergados et al. (2014) compared RO-derived observations of specific humidity with radiosondes and ERA-Interim under cloudy conditions in the tropics for August-October 2006. They used the direct method for computing specific humidity from the RO refractivity using the ERA-Interim temperatures. They found the following differences between zonal means of normalized RO and ERA-Interim observations of q (we computed the normalized differences from their data in Table

3 for the tropics):





Table 1: Normalized differences of zonal mean RO and ERA specific humidity in the tropics for cloudy conditions (computed from data in Vergados et al., 2014)

| Pressure (hPa) | VAR($\%^2$) | STD (%) |
|---|---|---|
| 925 | 320 | 17.8 |
| 850 | 460 | 21.4 |
| 700 | 1260 | 35.5 |
| 500 | 2860 | 53.5 |
| 400 | 4220 | 65.0 |
| 300 | 5625 | 75.0 |

10    The VAR values in Table 1 correspond to apparent errors (RO compared to ERA). As expected, they are larger than the estimated error variances for RO-Direct shown in Figure 6a because, as shown by Figure 5, the apparent errors are larger than the estimated true errors. This comparison indicates that the estimates of true errors in Figure 6a are reasonable.

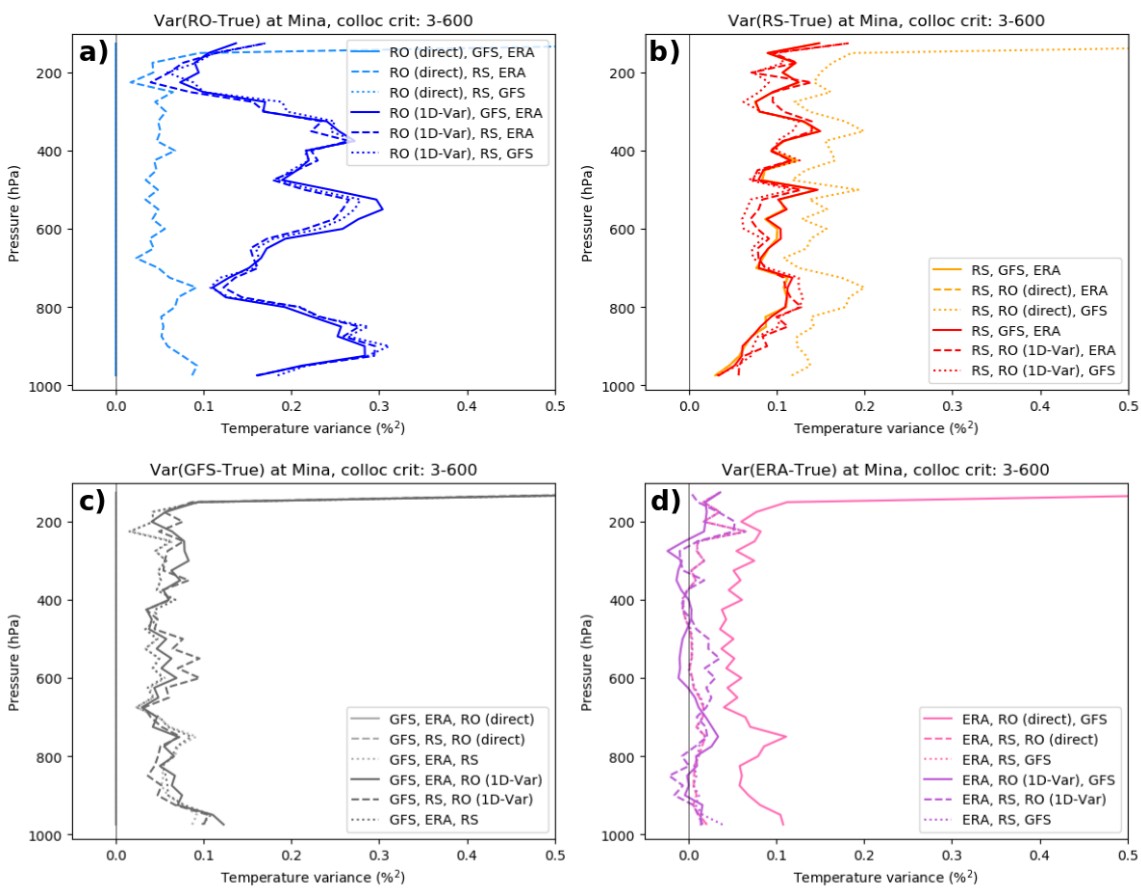

Figure 8: Estimated error variances (% squared) of temperature at Mina: (a) RO (b) RS, (c) GFS, and (d) ERA.

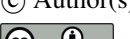



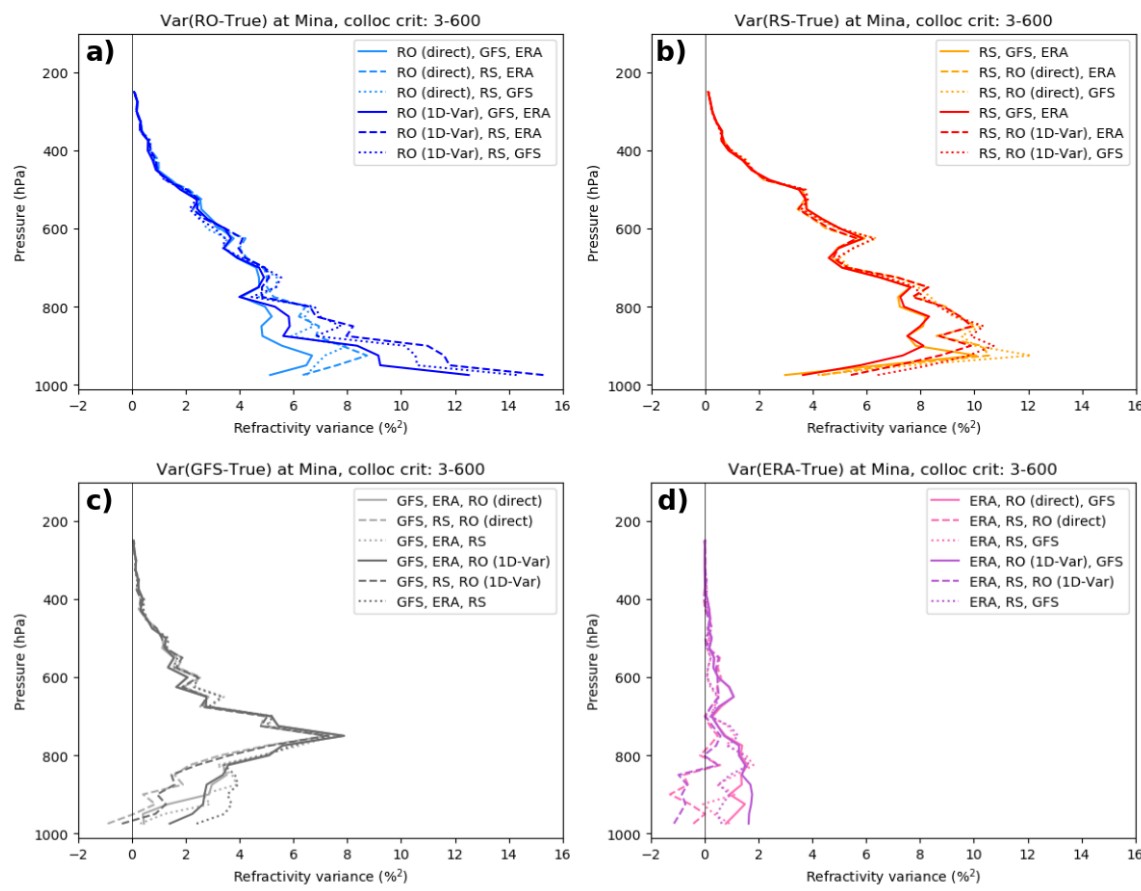

Figure 9: Estimated error variances (% squared) of refractivity at Mina. (a) RO, (b) RS, (c) GFS and (d) ERA.

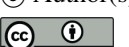



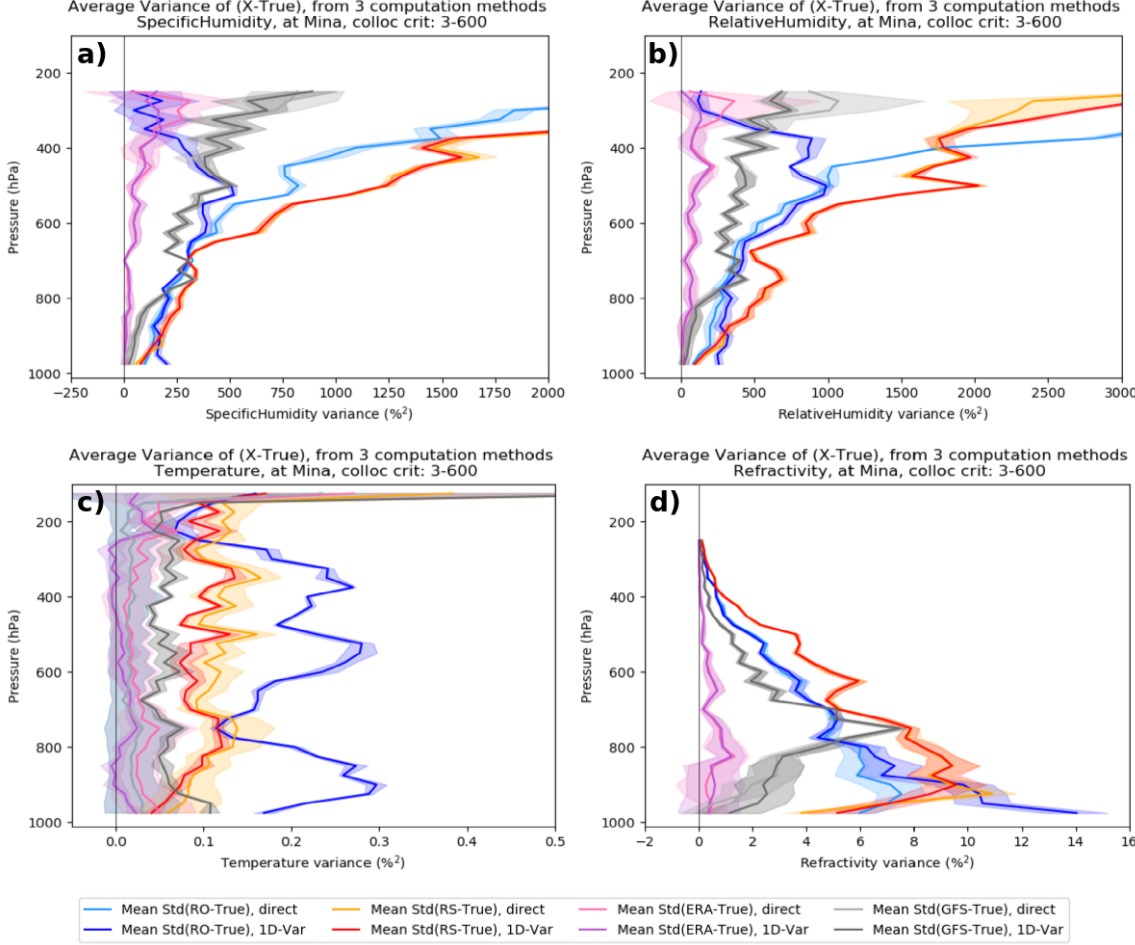

**Figure 10: Mean of the six estimates of error variance plots for q, RH, T and N using RO-Direct and RO 1D-VAR for each data set at Mina. The standard deviation about the mean is indicated by shaded areas. (a) specific humidity ; (b) relative humidity;(c) temperature; and (d) refractivity. RO (blue), radiosonde (red), GFS (gray), and ERA (purple).**



### 5.2 Summary of results at Naze, Ishi and Guam

The mean and STD error profiles for Naze, Ishi and Guam corresponding to the above results for Mina are presented in Appendix B. Here we summarize the main similarities and differences between the error variance estimates for these stations compared to those for Mina. In general, we find similar magnitudes and shapes of the profiles of the estimated error variances of the five data sets for all four variables (q, RH, T and N).

The estimated error profiles are especially similar for the three Japanese stations. This close similarity may be due primarily to the fact that the three locations are relatively close together and two of the three use the same type of radiosonde (Meisei).

The results from Guam are also similar in general magnitudes and shapes of the profiles to those from the three Japanese stations, but there are somewhat greater differences in some of the profiles (e.g. GFS q, RH and N and RS N). These differences 15   are likely due to the different location and the use of a different radiosonde type at Guam (VIZ/Sippican B2). The neglected error covariance terms are also likely different between the three Japanese stations, which are located in a data-rich region, and Guam, which is located in a data-sparse region. Thus the model errors are less likely to be highly correlated with a single observational system in the former than in the latter, where single observations may affect the models more significantly.

**6 Summary and discussion**

We used the "*N*-cornered hat" method to estimate vertical profiles of error variances of different observation and model data sets by computing the differences among the data sets at four fixed locations. We computed estimated error variances of four variables (specific humidity q, relative humidity RH, temperature T and refractivity N) for five data sets (ERA, GFS, 25   radiosondes (RS), RO-Direct and RO 1D-VAR) at four different locations in the tropics and subtropics for the year 2007. The stations are Guam, Ishigakijima, Minamidaitojima, and Naze. The latter three stations are on Japanese islands and are located quite close together (a few hundred km apart). We computed vertical profiles of estimated error variances for normalized differences from the 2007 ERA mean values of q, RH, T and N at the four stations using three linearly independent equations (Eq. (7)-(9)) neglecting all error covariance terms (COV). Ideally, with a very large sample of data pairs and zero correlation 30   of errors among the different data sets, all three equations would produce identical results. A finite data set and, more importantly, non-zero error correlations among the data sets, lead to three different estimates. The differences among the three estimates is a measure of these effects.



Although the neglect of the covariance terms affects the results to a noticeable degree in some of the estimated profiles, there is strong evidence that there is valid information in the estimated error profiles that rises above the noise caused by the neglect of the covariance terms and the limited data sample. This evidence is summarized as follows:

(1)   There is generally good agreement in the three estimated error profiles of the four variables for each of the five data sets at all four locations. It is unlikely that this agreement would occur by chance if the neglected error covariance terms were large enough to invalidate the results, because they would have to somehow combine or cancel in each of the three equations to give the observed similar results.

(2)   There are large differences in the overall structure (shape and magnitude) of the average vertical profiles of estimated error variances for the five data sets (Figure 10). These differences are significantly larger than the standard deviation of the differences among the three methods of estimating the error variances.

      (3)   The vertical variability, or scatter among the error estimates is similar at most height levels for specific humidity,
relative humidity and temperature. If the error covariance terms were significant, they would almost certainly vary with height, giving different agreement in estimated error profiles with height. For example, we know that RO is most accurate in the upper troposphere and least accurate in the lower troposphere. Also, the weight given RO in the models' data assimilation varies significantly with height, being largest in the upper troposphere and smallest in the lower troposphere. Thus one would expect the RO-ERA and RO-GFS error covariance terms to vary significantly
with height. Also, the RS errors as well as the ERA and GFS model errors vary with height. It is therefore unlikely that all of the neglected error covariance terms are the same at all heights.

      (4)   The general structure and magnitudes of the estimated error variance profiles are similar at the four locations. However, there are some small differences among the profiles at the four locations. In general, the vertical
variability or scatter, which is a measure of the effect of the neglected covariance terms as well as limited sample size, is smallest for Ishi, Naze and Mina and largest for Guam. Since the three Japanese stations are close together, this suggests that there is a difference in the error variance of the RS observations at these locations. There may also be small differences in the model errors over the Japanese stations, which are located in a data-rich area compared to Guam, which is located in a data-sparse region. The largest variability and largest error estimates occur at Guam
which uses a different radiosonde, which is thought to have large dry and wet water vapor biases due to sensor malfunctions (Vömel 2017, personal communication).

      (5)   The magnitudes of the estimated RO error variances are supported by previous published studies, including Kuo et al. (2004) and Chen et al. (2011).

      (6)   The estimated error profiles are smallest for the ERA-Interim model data set, which is a reasonable result since ERA uses an excellent model and data assimilation system, using many independent, quality checked observations. In fact, Vergados et al. (2015) state "ERA-Interim is one of the most advanced global atmospheric models simulating the state of the atmosphere with accuracy similar to what is theoretically possible (Simmons and
Hollingsworth, 2002) using a 4D-Var method (Simmons et al., 2005)."

      (7)   Our results show, in general, that the RO observations have smaller errors than the radiosonde errors, in agreement with previous studies.




*Code availability.* Code will be made available by the author upon request.

*Data availability.* Data can be made available from authors upon request.

## Appendix A: Derivation of estimates of error variances using four data sets and the *N*-cornered hat method

In this appendix we summarize the "*N*-cornered hat" method (Gray and Allan, 1974) for estimating error variances from *N* data sets. Variations and enhancements of the method have been applied to many diverse geophysical data sets, and for three data sets it is called the "three-cornered hat" method (Wriley, 2003) or "triple collocation method" (Stoffelen, 1998). Gray and Allan (1974) developed the method to estimate the absolute frequency stability of an ensemble of *N* clocks by forming all (*N*-1)(*N*-2)/2 triads under the assumption that the clock errors are uncorrelated. Each of the triads are the three-cornered hat (THC)

estimates. W.J. Wriley (2003) provides a summary of the TCH method and its history: http://www.wriley.com/3-CornHat.htm

The THC method has been used to estimate the stability of GNSS clocks using the measured frequencies from multiple clocks (Ekstrom and Koppang, 2006; Griggs et al., 2014, 2015; Luna et al., 2017). Valty et al. (2013) used the TCH method to estimate the geophysical load deformation computed from GRACE satellites, GPS vertical displacement measurements, and global

general circulation (GCM) models.

The "triple collocation method" was introduced by Stoffelen (1998), and has been widely used since in oceanography and hydrometeorology (e.g. Su et al., 2014; Gruber et al. 2015). It has been used to estimate the error variances of triplets of observation types to measure a diverse set of geophysical properties, including wave heights, sea surface temperatures,

precipitation, surface winds over the ocean, leaf-area index products, and soil moisture. For example, Stoffelen (1998) estimated the error variances of in-situ measurements, ERS scatterometer winds, and NCEP (National Centers for Environmental Prediction) forecast model wind speeds. Later, Vogelznang et al. (2011) compared four sets of scatterometer winds from ASCAT and SeaWinds with buoy measurements and ECMWF model forecasts of surface winds over the oceans to estimate the error variances and standard deviations of the different data sets and their combinations. Fang et al. (2012) used

Stoffelen's (1998) method to estimate the uncertainties in three different estimates of Leaf Area Index (LAI) products. McColl et al. (2014) extended the method by deriving a performance metric of the measurement system to the unknown truth, and applied the extended method to wind estimates from NWP, scatterometer and buoy wind estimates.





O'Carroll et al. (2008) compared three types of systems to measure sea-surface temperatures: two different radiometers and in situ observations from buoys. They discuss the assumption of the neglect of error correlations among the three data sets and the effect of representativeness errors. Roebeling et al. (2012) used the triple collocation method to estimate the errors

associated with three ways of estimating precipitation: the Spinning Enhanced Visible and Infrared Imager (SEVERI), weather radars, and ground-based precipitation rain gauges. They concluded that the method provides useful error estimates of these systems.

The major assumption in the above methods is that the errors of the three systems are uncorrelated. Correlations between any

or all of the three measurement systems will reduce the accuracy of the error estimates. Other factors that can reduce the accuracy of the error estimates include widely different errors associated with the three systems or a small sample size. These factors can lead to negative estimates of error variances, especially when the estimates are close to zero. All three of these factors potentially affect our estimations here, but the general agreement of the three methods suggests that the estimations are still reasonably valid and contain useful information.

In this appendix we summarize the derivation of the $N$-Cornered hat method as applied to four meteorological data sets, RO, RS, GFS and ERA. The error variance of a variable X (e.g. temperature, specific humidity, relative humidity, refractivity) is defined as $VAR(X) = [\sum(X-X_t)^2]/N$, where $X_t$ is the true (but unknown) value of X and the summation is over N samples. Let RO correspond to $X_{RO}$ or the value of X as estimated by RO, ERA correspond to $X_{ERA}$ or the value of X as estimated by ERA,

and similarly for GFS and RS (radiosondes). We then have

$$MS(RO-ERA) = VAR(RO) + VAR(ERA) - 2COV(RO,ERA) \qquad (1)$$

where MS(RO-ERA) is the mean square difference between RO and ERA and the last term is the error covariance between

RO and ERA.

In the estimation of the error variances for the four data sets, we assume that the RO errors and ERA errors are uncorrelated, so the error covariance term in Eq. (1) is zero, or in practice, negligibly small compared to the other terms). However, to show the complete (and exact) equations, we retain them here in the six equations involving the different pairs of data sets

$$MS(RO-GFS) = VAR (RO) + VAR(GFS) - 2COV(RO,GFS) \qquad (2)$$
$$MS(GFS-ERA) = VAR(GFS) + VAR(ERA) - 2COV(GFS,ERA) \qquad (3)$$
$$MS(RO-RS) = VAR(RO) + VAR(RS) - 2COV(RO,RS) \qquad (4)$$



MS(RS-ERA) = VAR(RS) + VAR(ERA) – 2COV(RS,ERA)  (5)

MS(RS-GFS) = VAR(RS) + VAR(GFS) – 2COV(RS,GFS)  (6)

It is possible to use these six equations to get six different estimates of the four unknowns VAR(RO), VAR(ERA), VAR(GFS)

5    and VAR(RS).  For example, the full six equations for computing VAR(RO) are:

2 VAR(RO) = MS(RO-ERA) + MS(RO-GFS) - MS(GFS-ERA)
        +2{COV(RO,ERA) + COV(RO,GFS) - COV(GFS,ERA)}  (7)

10   2 VAR(RO) = MS(RO-ERA) + MS(RO-RS) - MS(RS-ERA)
        +2{COV(RO,ERA) + COV(RO,RS) – COV(RS,ERA)}  (8)

2 VAR(RO) = MS(RO-GFS) + MS(RO-RS) - MS(RS-GFS)
        +2{COV(RO,GFS) + COV(RO,RS) – COV(RS,GFS)}  (9)

O'Carroll et al. (2008) present these equations for a system of three observation types (their Eq. (1)). Additionally, VAR(ERA),

VAR(GFS), and VAR(RS) can be computed separately from different combinations of Eqs. (1)-(6), and these values can be

substituted into Eqs. (1), (2), and (4) to compute the remaining three estimates of VAR(RO):

20   VAR(RO) =  MS(RO-ERA) + 2COV(RO,ERA) - VAR (ERA)  (10)

VAR(RO) =  MS(RO-GFS) + 2COV(RO,GFS) - VAR (GFS)  (11)

VAR(RO) =  MS(RO-RS) + 2COV(RO,RS) - VAR (RS),  (12)

where VAR(ERA) is computed from Eqs. (1), (2) and (3)

VAR(ERA) = 1/2 [MS(GFS-ERA)+MS(RS-ERA)-MS(RS-GFS)]
        +COV(GFS,ERA)+COV(RS,ERA)-COV(RS,GFS),  (13)

VAR(GFS) is computed from Eqs. (1), (4) and (5)

VAR(GFS) = ½[MS(GFS-ERA) + MS(RS-GFS) –MS(RS-ERA)]
            +COV(GFS,ERA) +COV(RS,GFS) –COV(RS,ERA)  (14)





and VAR(RS) is computed for Eqs. (2), (4), and (6)

$$VAR(RS) = \tfrac{1}{2}[MS(RS\text{-}ERA) + MS(RS\text{-}GFS) - MS(GFS\text{-}ERA)]$$
$$+COV(RS,ERA) + COV(RS,GFS) - COV(GFS,ERA). \qquad (15)$$

The same procedure can be used to derive six equations for estimating the error variances for the other three data sets, RS, ERA, and GFS (not shown).

So for each of the data sets RO, RS, ERA, and GFS, there are six different ways to estimate the respective error variances. The first three of these equations are linearly independent; these are the three triads in the Gray and Alan (1974) N-cornered hat method, and the other three are linearly dependent, but are different ways of combining the observed data sets to estimate the error covariances.

If all the neglected error COV terms were in fact identically zero, the set of observations in the pairs (RO,ERA), (RO,GFS), (GFS,ERA), (RO,RS), (RS,ERA) and (RS,GFS) are the same (they are in our case), and the sample size was very large (much larger than our sample size), all six estimates of the error variances would be the same. The fact that they give different solutions is because the neglected COV terms in are in realty not zero, and hence their neglect affects the six approximate equations in different ways to give six different solutions. The relatively small sample size N also contributes to the differences

in the six solutions, which are a measure of these effects.

Figures A1-A4 compare the mean of the estimated standard deviations of the errors (computed from the square root of the estimated error variances) associated with RO 1D-VAR, RS, ERA and GFS for specific humidity, relative humidity, temperature and refractivity computed from the six equations (left) and three equations (right) for Guam and Mina.





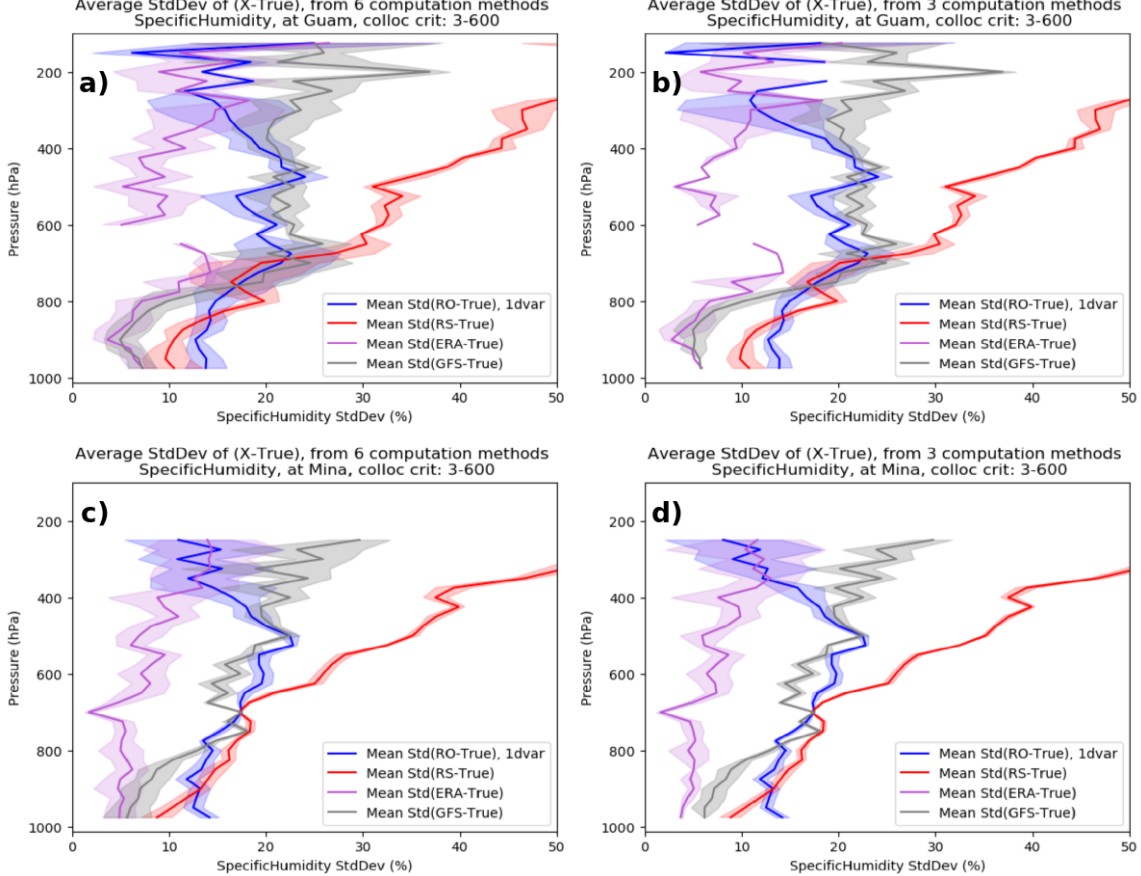

**Fig. A1:** Mean of six and three error estimates of normalized specific humidity at Guam (top) and Mina (bottom). The standard deviation about the means is shown by the shading.





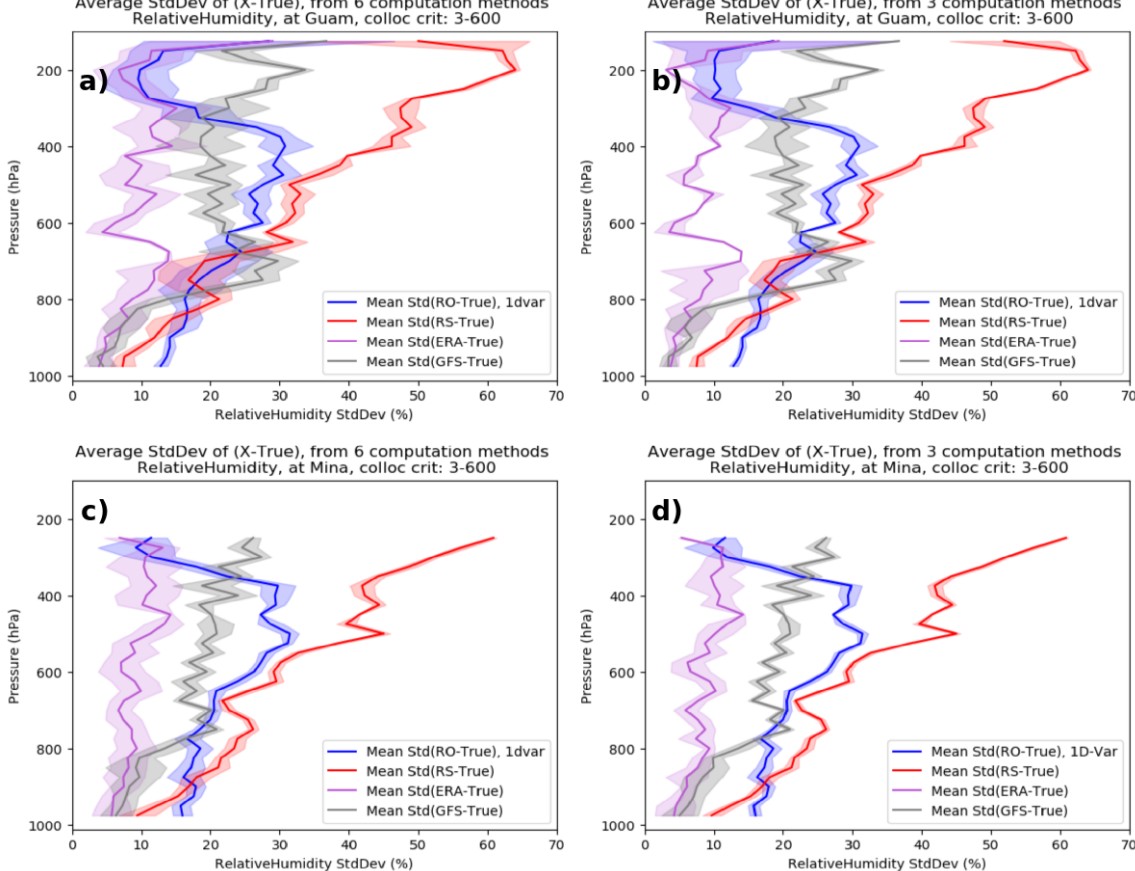

**Fig. A2: Mean of six and three error estimates of normalized relative humidity at Guam (top) and Mina (bottom). The standard deviation about the means is shown by the shading.**





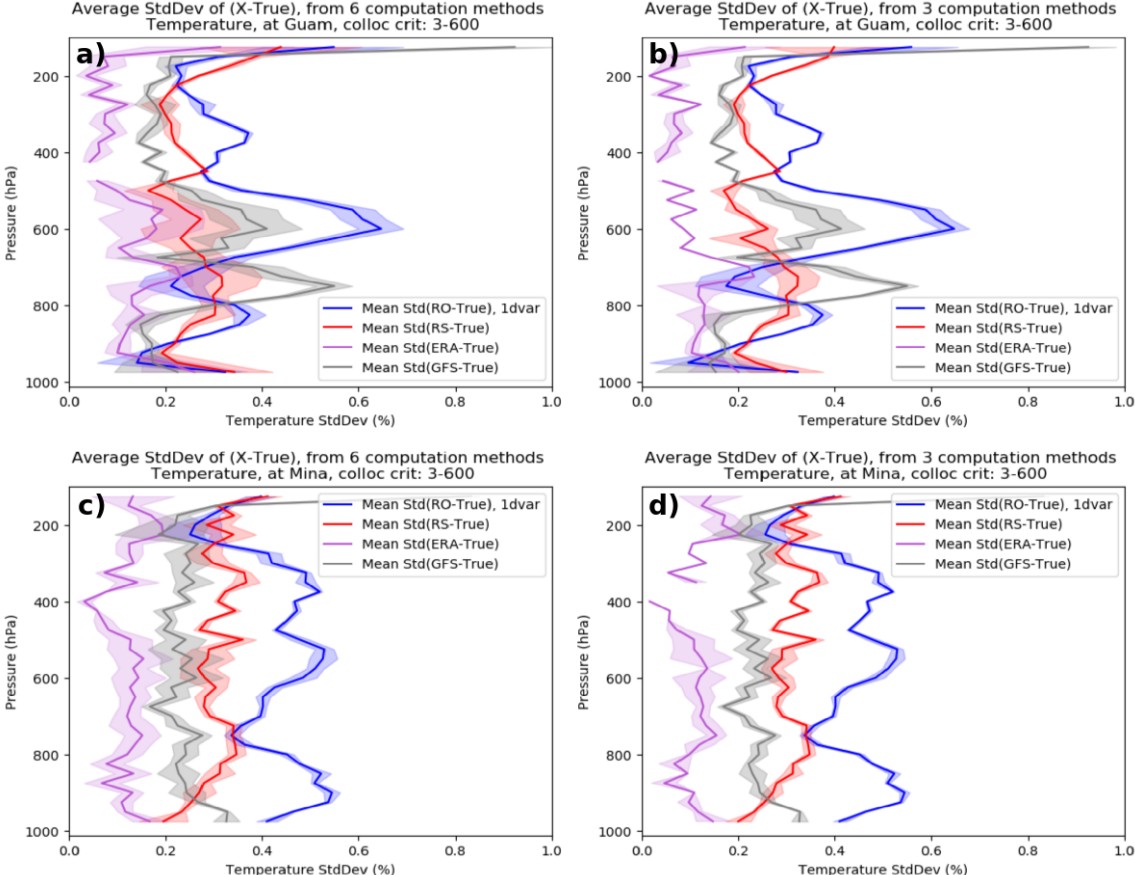

**Fig. A3: Mean of six and three error estimates of normalized temperature at Guam (top) and Mina (bottom). The standard deviation about the means is shown by the shading.**





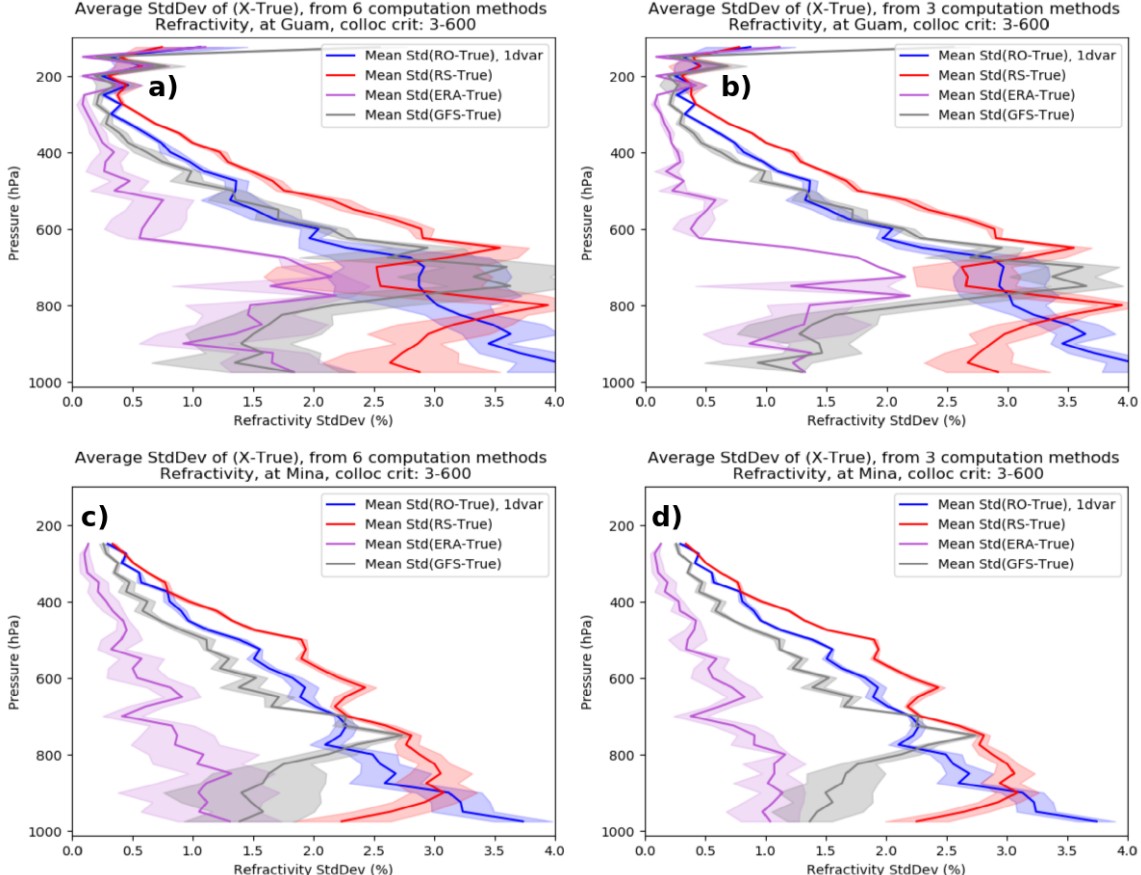

**Fig. A4: Mean of six and three error estimates of normalized specific humidity at Guam (top) and Mina (bottom). The standard deviation about the means is shown by the shading.**

There is generally good agreement among all the estimated error profiles of the four observables for each of the data sets at all four locations. The mean profiles are quite similar, but the scatter (STD) is smaller for the three independent solutions. It is unlikely that this agreement would occur by chance if the neglected error covariance terms were large enough to invalidate the results, because they would have to somehow combine or cancel in each of the three equations to give the observed similar results. The error covariance terms include error correlations between

RO and ERA

RO and GFS





GFS and ERA

RO and RS

RS and ERA

RS and GFS

These combine in different ways in the six equations; the neglected terms in each equation are:

Eq. (7):        COV(RO,ERA)+COV(RO,GFS)-COV(GFS,ERA)

Eq. (8):        COV(RO,ERA)+COV(RO,RS) – COV(RS,ERA)

Eq. (9):        COV(RO,GFS)+COV(RO,RS) – COV(RS,GFS)

Eq. (10):       2COV(RO,ERA) - {COV(GFS,ERA)+COV(RS,ERA)-COV(RS,GFS)}

Eq. (11):       2COV(RO,GFS) - {COV(GFS,ERA) +COV(RS,GFS) –COV(RS,ERA}

Eq. (12):       2COV(RO,RS) – {COV(RS,ERA) + COV(RS,GFS) – COV(GFS,ERA)}

Since in all cases the results are similar, these six combinations must be approximately the same and most likely smaller that the terms involving the mean squares of the difference between the various data sets used to compute the estimates.





## Appendix B: Mean and standard deviations of three independent error estimates of q, RH, T, and N using RO Direct and RO 1D-VAR at Guam, Ishi, Mina and Naze.

**Figure B1: Mean and standard deviations (shading) of the three estimates of normalized specific humidity using RO Direct and RO 1D-VAR at (a) Guam, (b) Ishi, (c) Mina and (d) Naze.**





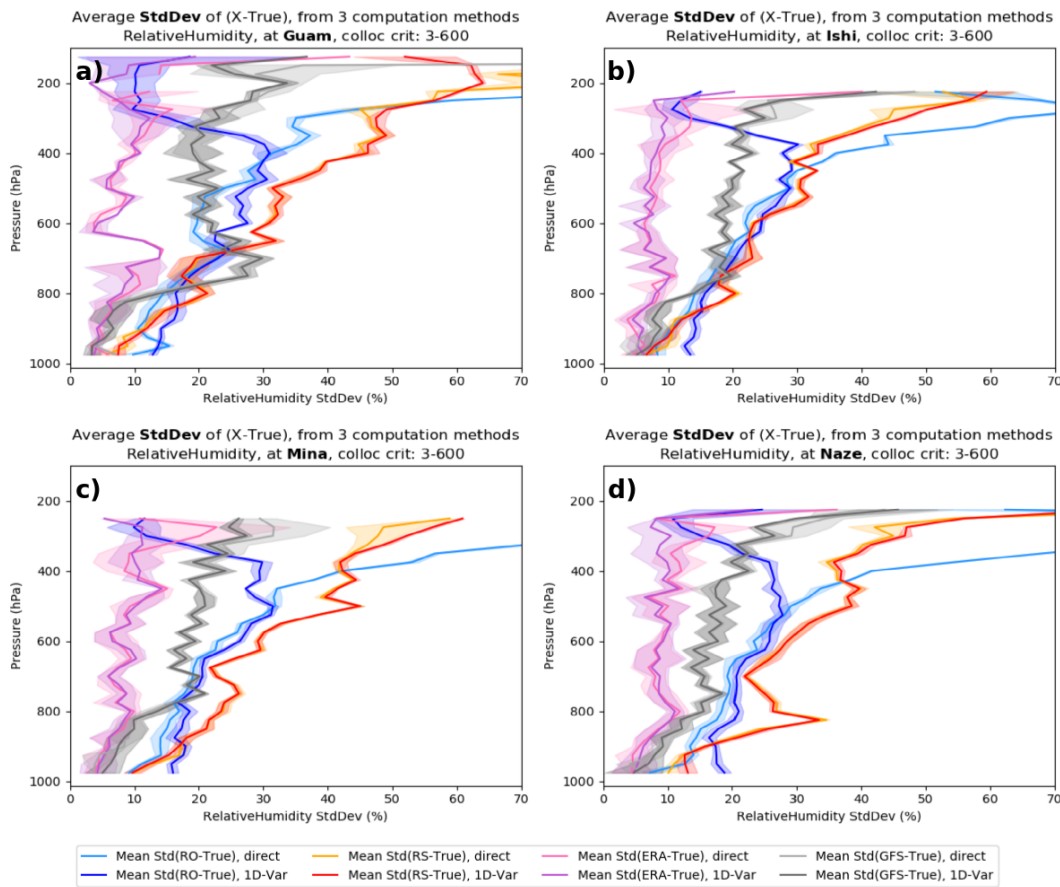

**Figure B2: Mean and standard deviations (shading) of the three estimates of normalized relative humidity using RO Direct and RO 1D-VAR at (a) Guam, (b) Ishi, (c) Mina and (d) Naze.**

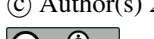



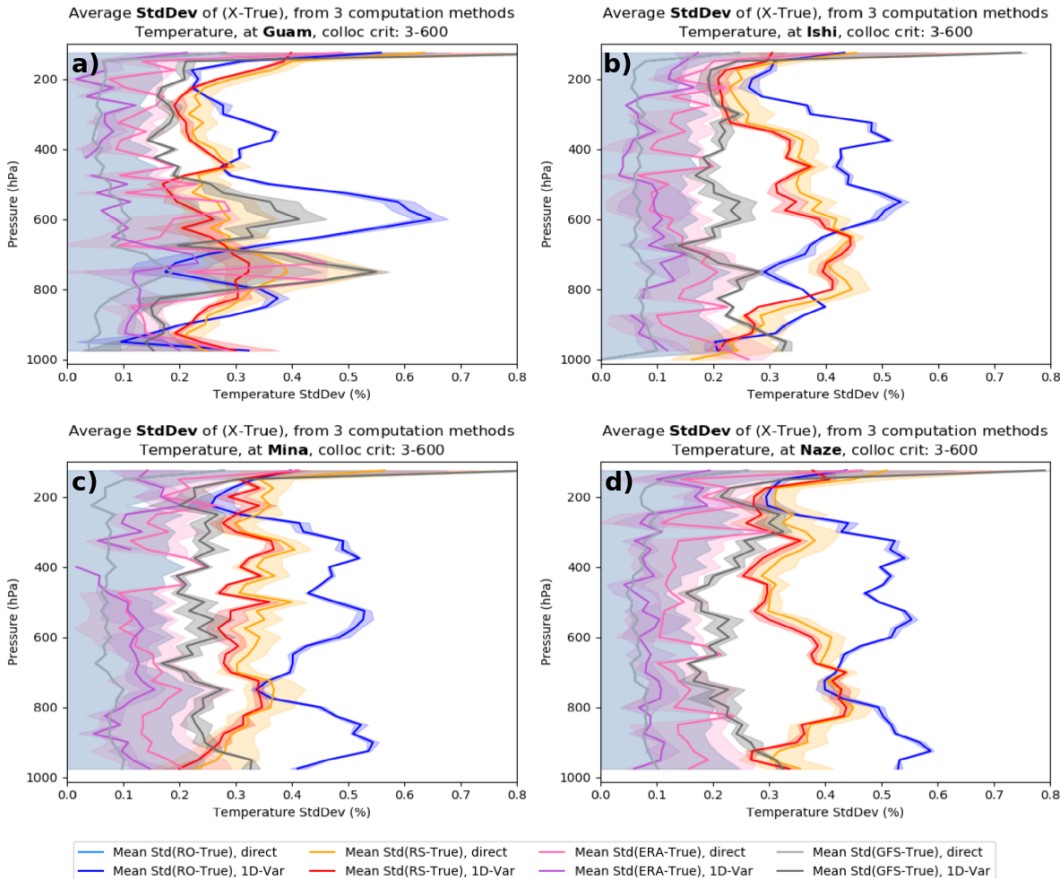

**Figure B3: Mean and standard deviations (shading) of the three estimates of normalized temperature using RO Direct and RO 1D-VAR at (a) Guam, (b) Ishi, (c) Mina and (d) Naze.**





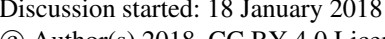

**Figure B4: Mean and standard deviations (shading) of the three estimates of normalized refractivity using RO Direct and RO 1D-VAR at (a) Guam, (b) Ishi, (c) Mina and (d) Naze.**

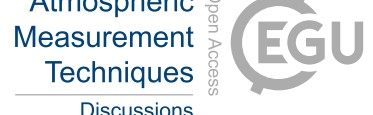

Gilpin, S., Rieckh, T., and Anthes, R.: Reducing representativeness and sampling errors in radio occultation-radiosonde comparisons, in preparation, 2017.

Gray, J.E. and Allan, D.W.: A method for estimating the frequency stability of an individual oscillator, Proc. of 28[th] Frequency
Control Symp., 29-31 May 1974, Atlantic City, New Jersey, pp. 243-246, 1974.

Griggs, E., Kursinski, E. R., and Akos, D.: An investigation of GNSS atomic clock behaviour at short time intervals, GPS Solutions, 18:443-452, doi 10.1007/s10291-013-0343-7, 2014.

Griggs, E., Kursinski, E., and Akos, D.: Short-term GNSS satellite clock stability, Radio Sci., 50, 813–826, https://doi.org/10.1002/2015RS005667, 2015.

Gruber, A., Su, C.-H., Zwieback, S., Crow, W., Dorigo, W., and Wagner, W.: Recent advances in (soil moisture) triple collocation analysis, Int. J. Appl. Earth Observations and Geoinformation, 45, Part B, 200-211,
https://doi.org/10.1016/j.jag.2015.09.002, 2016.

Ho, S.-P., Zhou, X., Kuo, Y.-H., Hunt, D., and Wang, J.-H.: Global evaluation of radiosonde water vapor systematic biases using GPS radio occultation from COSMIC and ECMWF analysis, Remote Sensing, 2, 1320–1330, https://doi.org/10.3390/RS2051320, 2010.

Ho, S.-P., Peng, L., and Vömel, H.: Characterization of the long-term radiosonde temperature biases in the upper troposphere and lower stratosphere using COSMIC and Metop-A/GRAS data from 2006 to 2014, ACP, 17, 4493–4511, https://doi.org/10.5194/acp-17-4493-2017, 2017.

Hollingsworth, A. and Lönnberg, P.: The statistical structure of short-range forecast errors as determined from radiosonde data. Part I: The wind field, Tellus A, 38A, https://doi.org/10.1111/j.1600-0870.1986.tb00460.x, 1986.

Kleist, D. T., Parrish, D. F., Derber, J. C., Treadon, R., Wu, W.-S., and Lord, S.: Introduction of the GSI into the NCEP Global Data Assimilation System, Wea. Forecasting, 24, 1691–1705, https://doi.org/10.1175/2009WAF2222201.1, 2009.

Kuo, Y.-H., Wee, T.-K., Sokolovskiy, S., Rocken, C., Schreiner, W., Hunt, D., and Anthes, R. A.: Inversion and error estimation of GPS radio occultation data, J. Meteor. Soc. Japan, 82, 507–531, 2004.





Kursinski, E. R., Hajj, G. A., Bertiger, W. I., Leroy, S. S., Meehan, T. K., Romans, L. J., Schofield, J. T., McCleese, D., Melbourne, W., Thournton, C., Yunck, T., Eyre, J., and Nagatani, R.: Initial results of radio occultation observations of Earth's atmosphere using the Global Positioning System, Science, 271, 1107–1110, https://doi.org/10.1126/science.271.5252.1107, 1996.

Ladstädter, F., Steiner, A. K., Schwärz, M., and Kirchengast, G.: Climate intercomparison of GPS radio occultation, RS90/92 radiosondes and GRUAN from 2002 to 2013, Atmos. Meas. Tech., 8, 1819–1834, https://doi.org/10.5194/amt-8-1819-2015, 2015.

Luna, D., Pérez, D., Cifuentes, A., and Gómez, D.: Three-Cornered Hat Method via GPS Common-View Comparisons, IEEE
Trans. Instrument. Measure., 66, 2143–2147, https://doi.org/10.1109/TIM.2017.2684918, 2017.

McColl, K.A., Vogelzang, J., Konings, A.G., Entekhabi, D., Piles, M., and Stoffelen, A.: Extended triple collocation: Estimating errors and correlation coefficients with respect to an unknown target, Geophys. Res. Letters, 41, 6229-6236, doi:10.1002/2014GL061322, 2014.

O'Carroll, A.G., Eyre, J.R., and Saunders, R.S: Three-way error analysis between AATSR, AMSR-E, and in situ sea surface temperature observations. J. Atmos. Oceanic Tech., 25,1197-1207, https://doi.org/10.1175/2007JTECHO542.1, 2008.

Parrish, D. F. and Derber, J. C.: The National Meteorological Center's Spectral Statistical-Interpolation analysis system, Mon.
Wea. Rev., 120, 1747–1763, https://doi.org/10.1175/1520-0493(1992)120<1747:TNMCSS>2.0.CO;2, 1992.

Rieckh, T., Anthes, R., Randel, W., Ho, S.-P., and Foelsche, U.: Evaluating tropospheric humidity from GPS radio occultation, radiosonde, and AIRS from high-resolution time series, Atmos. Meas. Tech. Discuss., submitted, 2017.

Roebeling, R.A., Wolters, E. L. A., Meirink, J.F. and Leijnse, H.: Triple collocation of summer precipitation retrievals from SEVIRI over Europe with gridded rain gauge and weather radar data, J. Hydrometeor, 13, 1552-1566, doi:10.1175/JHM-D-11-089.1, 2012.

Su, C.-H., Ryu, D., Crow, W.T., and Western, A.W.: Beyond triple collocation: Applications to soil moisture monitoring, J.
Geophys. Res. Atmospheres, 119, 6419-6439, doi:10.1002/2013JD021043, 2014.

Simmons, A. J. and Hollingsworth, A.: Some aspects of the improvement in skill of numerical prediction, Quart. J. Roy. Meteor. Soc., 128, 647–677, https://doi.org/10.1256/003590002321042135, 2002.



Simmons, A. J., Hortal, M., Kelly, G., McNally, A., Untch, A., and Uppala, S.: ECMWF Analyses and Forecasts of Stratospheric Winter Polar Vortex Breakup: September 2002 in the Southern Hemisphere and Related Events, J. Atmos. Sci., 62, 668–689, https://doi.org/10.1175/JAS-3322.1, 2005.

Smith, E. and Weintraub, S.: The constants in the equation for atmospheric refractive index at radio frequencies, Proc. IRE,

41, 1035–1037, 1953.

Stoffelen, A.: Toward the true near-surface wind speed: Error modeling and calibration using triple collocation, J. Geophys. Res. Oceans, 103, 7755-7766, doi:10.1029/97JC03180, 1998.

Valty, P., de Viron, O., Panet, I., Camp, M. V., and Legrand, J.: Assessing the precision in loading estimates by geodetic techniques in Southern Europe, Geophys. J. Int., 194, 1441–1454, https://doi.org/https://doi.org/10.1093/gji/ggt173, 2013.

Vergados, P., Mannucci, A. J., and Ao, C. O.: Assessing the performance of GPS radio occultation measurements in retrieving tropospheric humidity in cloudiness: A comparison study with radiosondes, ERA-Interim, and AIRS data sets, J. Geophys.

Res., 119, 7718–7731, https://doi.org/10.1002/2013JD021398, 2014.

Vergados, P., Mannucci, A. J., Ao, C. O., Jiang, J., and Su, H.: On the comparisons of tropical relative humidity in the lower and middle troposphere among COSMIC radio occultations and MERRA and ECMWF data sets, Atmos. Meas. Tech., 8, 1789–1797, https://doi.org/10.5194/amt-8-1789-2015, 2015.

Vogelzang, J., Stoffelen, A., Verhoef, A., and Figa-Saldaña: On the quality of high-resolution scatterometer winds, J. Geophy. Res., 116, C10033, doi:10.1029/2010JC006640, 2011.

