# Peer review of "Estimating observation and model error variances using multiple data sets"

_Atmospheric Measurement Techniques, 2017_

## Referee Comment (RC1) · Anonymous Referee #1 · 26 Feb 2018

Manuscript Number: amt-2017-487

Title: Estimating observation and model error variances using multiple data sets

Author: Richard A. Anthes and Therese Rieckh

**General Comments:**

The authors estimate the observation and model error variances by using the "N-cornered hat method". In this study, they estimate the error variances for observation and model in several variables, e.g., refractivity, temperature, specific humidity and relative humidity. They compare their results with previous studies and find that the error patterns are consistent. The errors characters for the GPS RO retrieved temperature and moisture are rarely discussed in previous studies, and it is good to see the

estimation in the study. For the manuscript, I have some comments as follows.

**Specific comments:**

1. The manuscript discusses the observational error variances for the refractivity, temperature, and moisture (q and RH) from GPS RO, but not for the bending angle. The bending angle from GPSRO has been assimilated in several operational centers for weather forecast. Is it possible to provide the error estimation for the bending angle as well? This could be interesting and useful for community users and/or the NWP people.

2. The samples are picked up within 600km and 3h for comparison, is the criteria the same for ERA and GFS? For example, do the authors apply a spatiotemporal interpolation when comparing ERA and RO? If it uses the co-location criteria, how much could this affect the error variance?

3. The abstract does not point out the major conclusion of the study. I would suggest to add the part into the abstract.

4. The statistics are based on samplings near the four stations. According to previous studies, the observational error variance could vary with latitudes, can the results in this study be applied globally?

5. In Fig. 5a, A1 and A3, there are several gaps in STD (ERA-True). The ERA-Interim data should be continuous.

6. In the manuscript, the "N" can be represented for several meanings, e.g., RO refractivity, the number of samples for statistic, and number of data sets, etc. It would be better to use different characters to avoid confusing.

7. Page 4 line 15: There are three types of COSMIC data provided from CDAAC, i.e., re-processed, post-processed, and real-time data. In this manuscript, it uses the ERA-Interim data as the background for the 1DVAR retrieval, do the authors get the COSMIC data from re-processed data?

**Technical corrections:**

1. The variables should be in the same form with italics, for example, page 4 line 17: "Specific humidity q" → please change to "Specific humidity $q$"; page 4 line 18: "water vapor pressure e" → "water vapor pressure $e$".

2. Page 22 line 14 and line 16: three-cornered hat (THC) → three-cornered hat (TCH)

3. Pages 24 and 25: The descriptions for equations (13)-(15) are inconsistent, for example, the equation (13) is computed from Eqs. (3), (5) and (6), not (1), (2) and (3) that indicated in line 26.

4. Page 29 Fig. A4 figure caption: Mean of six . . .of normalized "specific humidity" → should be "refractivity"
* * *

---

## Referee Comment (RC2) · Anonymous Referee #2 · 19 Apr 2018

This manuscript applies the N-cornered hat technique to estimate errors in geophysical measurements: radio occultation, radiosondes, ERA-Interim reanalysis and weather forecast outputs at four locations in the tropics and subtropics. The N-cornered hat technique is closely related to the method of triple collocation, which has been widely applied to geophysical datasets in the literature (as the authors note). However, some subtle differences between the approaches are missed in this analysis, which may impact the results.

One important difference between triple collocation (TC) and the three-cornered hat (3CH) is the treatment of the underlying truth. TC treats the underlying truth as a random variable (Stoffelen 1998), whereas 3CH does not. As a result, TC requires an additional assumption compared to 3CH: the errors must be uncorrelated with the un-

derlying truth. Since the underlying truth is not considered to be a random variable in 3CH, the correlation between the errors and truth is always zero. So, should the underlying truth be treated as a random variable (as in TC) or as deterministic (as in 3CH)? I would argue that, for assessing the stability of clocks, the assumption of a deterministic underlying truth is quite reasonable. However, when considering atmospheric applications, as in this study, it is hard to justify. The atmosphere is a chaotic system with substantial internal variability. Differences between measurements can be due to measurement errors, but they can also be due to the internal variability of the chaotic system. In Figure 3a, for example, there are clear differences between the specific humidity profile estimated by GFS and RS on 13 January, 2007 at 00:23 UTC. Even if the measurement and modelling errors of both GFS and RS were both zero, we would expect there to be some difference between these two profiles because of the internal variability of the system, even accounting for some assimilation of observations into the GFS. Yet the 3CH implicitly attributes ALL differences between different measurement types to measurement errors in one or more measurement types. This seems misguided and is one of the reasons TC is typically applied to geophysical measurements rather than 3CH. Treating the underlying truth as deterministic rather than random leads the authors to neglect the impacts of possible covariance between the errors and the underlying truth, which is likely biasing the error estimates in this study.

A second important difference is that TC accounts for multiplicative biases in the measurements in a way that 3CH does not. 3CH implies the following measurement model:

$i = T + e\_i \ldots$ (1)

where $T$ is the unknown true value, and $e\_i$ is the error in measurement $i$. In contrast, triple collocation starts with the measurement model:

$i = a\_i + b\_i*T + e\_i \ldots$ (2)

where $a\_i$ and $b\_i$ are additive and multiplicative biases, respectively (see Gruber et al. (2016), equation 1). The advantage of the measurement model in TC is that it is

robust to multiplicative biases. If multiplicative biases exist but are not factored into the measurement model (as in 3CH), the multiplicative biases will lead to substantial correlations between the errors and the unknown truth. In turn, the neglect of these correlations will bias 3CH error estimates.

Given both of these problems with the analysis, it is not surprising that the estimated errors (for example, in Figure 10) are not internally consistent (as they should be, if either technique – 3CH or TC – were applied appropriately). While 3CH and TC are similar in many respects, there are good reasons to use TC rather than 3CH when characterizing errors in geophysical data. Therefore, to address these concerns, I recommend the authors reframe their analysis using TC rather than 3CH. The differences between the techniques are relatively small, but significant, and warrant a substantial rewrite of the manuscript, in my view.

Specific comments Line 7, p 16: "...indicate that our estimates are reasonable and consistent with these studies." It seems unreasonable to be making this claim given the estimates in this study vary enormously. It is easy for an imprecise estimate to be consistent with previous studies, but this is not particularly informative.

References

Stoffelen, Ad. "Toward the True Near-Surface Wind Speed: Error Modeling and Calibration Using Triple Collocation." Journal of Geophysical Research 103, no. C4 (1998): 7755–66.

Gruber, A., C. -H. Su, S. Zwieback, W. Crow, W. Dorigo, and W. Wagner. "Recent Advances in (Soil Moisture) Triple Collocation Analysis." International Journal of Applied Earth Observation and Geoinformation, Advances in the Validation and Application of Remotely Sensed Soil Moisture - Part 1, 45, Part B (March 2016): 200–211. https://doi.org/10.1016/j.jag.2015.09.002.

---

## Referee Comment (RC3) · Anonymous Referee #3 · 1 May 2018

General comments:

This is a very well written paper describing how one can estimate the error variances of different datasets of atmospheric profiles using the differences between three or more independent datasets of the same variable. The method ("N-cornered hat") is used to estimate the error variances in tropospheric profiles of four variables in five datasets at four locations in Japan and in Guam. The results indicate that the main assumption of neglecting error correlations between the datasets is reasonably valid.

I enjoyed reading the paper and have only a few minor specific comments and technical corrections.

Specific comments:

[Figure]

Page 3, line 4: "Paper 1" is only referred to once a few lines below (line 9). Thus it seems overkill to introduce it as "Paper 1" here. Perhaps in line 9 you can just say something like "Because the focus in Rieckh et al. (2017) was ...".

Page 3, section 2.1. Please clarify if you are using analysis or forecast products from ERA-Interim. If you are using forecast products, that would be another good argument for small correlations, since the ERA forecasts only contain earlier observations via the assimilation, and are therefore independent of the observations that they are compared to.

Figs. 6-9: The results without RO (those involving only RS,GFS,ERA) are listed twice in b,c,d panels. I would have expected identical curves, but there are small differences (e.g. refractivity from GFS,ERA,RS below 900 hPa in Fig 9c - light and dark gray dotted curves). Why is that?

Page 15, line 26: "... mean of the six estimates of the error variances ...". But it looks like the light and dark results are separated. Is the mean only over three estimates each? How is the standard deviation with only three estimates taken? For small samples, it becomes important that the denominator in the expression for the sample variance is more correctly written as N-1 (N=3 here). This would also alter the shaded areas in the appendix.

Page 21, line 30: "... dry and wet water vapor biases ...". Needs reformulation.

Page 24, line 26: Shouldn't it be Eqs. (3), (5), and (6)? And the same for the following eqs.

However, eqs. (10), (11), and (12), with (13), (14), and (15) inserted are not additional independent equations (as you also write). For example, (10) with (13) inserted is equivalent to (7) + (8) - (9). In general, you could get many more (infinitely many) equations if you don't care that they are dependent, namely A*(7)+B*(8)+C*(9), where A, B, and C are any numbers, except those where A+B+C=0. Thus, there are not only

six different ways. There are three independent ways and infinitely many if you also count dependent ones that can be formed by linear combinations of the first three. I think you need to make clear that there is nothing unique about the three additional equations that you choose. As it is, one could get the impression that they are in some way special (in line 5 you write "the full six equations" and in line 18 "the remaining three estimates"). Perhaps they are special if one can show that they give rise to the smallest possible standard deviation of the variance estimates. I don't know if that is the case. In the first three independent equations, there are 3/2 MS terms, and 3 COV terms involved. In the additional three equations there are 5/2 MS terms and 5 COV terms. The more terms there are, the larger the standard deviation of the variance estimates will become, at least potentially. Interesting stuff!

Page 25, line 15-16: I do not understand this part of the sentence: "...the set of observations in the pairs (RO,ERA), (RO,GFS), (GFS,ERA), (RO,RS), (RS,ERA) and (RS,GFS) are the same (they are in our case)..." What are the set of observations here?

Technical corrections:

Page 2, line 3: model -> modeled

Page 3, line 14: Missing "a" in front of global.

Page 11, line 6: Acronym STD should be written out first time.

Page 12, line 32: that -> than

Page 15-16, lines 30,1-2: Should RS and RO be switched in the text here? RO is the one that oscillates between 0.1 and 0.3 (% squared). RS is fairly constant about 0.1.

Page 21, line 17: Missing "to" in front of RO.

Page 22, line 27: Vogelznang -> Vogelzang

Page 25, line 18: "in are in"?

Page 29, line 6: "... four locations?" There are only results for two here, Guam and Mina.

Page 30, line 20: that -> than

---

## Author Response (AR1)

**Specific comments:**
*1. The manuscript discusses the observational error variances for the refractivity, temperature, and moisture (q and RH) from GPS RO, but not for the bending angle. The bending angle from GPSRO has been assimilated in several operational centers for weather forecast. Is it possible to provide the error estimation for the bending angle as well? This could be interesting and useful for community users and/or the NWP people.*

We agree that the bending angle (BA) error variance estimates would be very interesting. However, as shown by Gilpin et al. (2018b), adding BA to the observations currently in the study (refractivity, temperature, relative humidity and specific humidity) is not trivial. To use the 3CH method, BA corresponding to radiosonde, ERA-Interim and GFS data sets would have to be computed from the refractivity of these data sets using a forward model (the Abel Integral). The Abel Integral requires vertical differentiation of refractivity, which requires accurate vertical interpolation of refractivity between the model (or radiosonde) levels. The vertical resolutions of the models (and often also the radiosonde) are low compared to the resolution required to accurately approximate the Abel Integral. For these low-resolution data sets, small-scale atmospheric structures as well as the quasi-exponential variation of refractivity with height, can create large errors in the calculated bending angle. Thus, to add discussion on the forward model and interpolation methods and then to compute error variances for bending angles is outside the scope of this study.

*2. The samples are picked up within 600km and 3h for comparison, is the criteria the same for ERA and GFS? For example, do the authors apply a spatiotemporal interpolation when comparing ERA and RO? If it uses the co-location criteria, how much could this affect the error variance?*

In response to this comment, we have added more detail to how the data sets are co-located. We have added a new Section 2.5 (on Page 4) titled "Co-location of the data sets":

"The locations of the four radiosonde stations are chosen for the comparisons. We use RO observations that are located within 600 km and 3 hours of the radiosonde launches. CDAAC provides GFS and ERA profiles that are already linearly interpolated in space and time to the RO location and time. These interpolated profiles, along with the RO observations, were corrected for their time and spatial differences from the radiosonde data using a model correction algorithm (Gilpin et al., 2018). Thus the effect of spatial and temporal differences among the data sets is expected to be minor."

*3. The abstract does not point out the major conclusion of the study. I would suggest to add the part into the abstract.*

We agree and have added the following paragraph to the abstract:

"Our results show that different combinations of the four data sets yield similar relative and specific humidity, temperature, and refractivity error variance profiles at the four stations, and these estimates are consistent with previous estimates where available. These results thus indicate that the correlations of the errors among all data sets are small and the 3CH method yields realistic error variance profiles. The estimated error variances of the ERA-Interim data set are smallest, a reasonable result considering the excellent model and data assimilation system and assimilation of high-quality observations. For the four locations studied, RO has smaller error variances than radiosondes, in agreement with previous studies."

*4. The statistics are based on samplings near the four stations. According to previous studies, the observational error variance could vary with latitudes, can the results in this study be applied globally?*

The main point of this paper is to show how the 3CH method can be used to estimate vertical profiles of temperature, relative and specific humidity, and refractivity error variances of five data sets at selected challenging locations in the tropics and sub-tropics. The error variances of all data sets are likely to vary with latitude, and also over different regions and seasons (at least for the models and radiosondes with their different radiosonde sensor types). A study using different locations, seasons, and years would be interesting and would be a better indicator of the error variances on a global scale. A calculation of these error variances at many locations over a number of years would also be interesting, to see how the errors of the various systems vary over time.

*5. In Fig. 5a, A1 and A3, there are several gaps in STD (ERA-True). The ERA-Interim data should be continuous.*

The ERA data themselves are continuous, but the estimated ERA-Interim error variances are close to zero. Furthermore, we neglect the error covariance terms in the computation and have a limited sample size. The combination of these three factors can lead to negative estimated error variances at times when the true error variance is so close to zero, and the STD is thus undefined. This is discussed briefly at the end of Section 3 and also in Appendix A, with references.

We added the following sentence on page 9 lines 5-7:
"The gap in the computed ERA error STD in Fig. 5a occurs due to negative estimated error variance values, which can result from having a limited sample size, neglecting error covariance terms during computation, and having an error variance that is already close to zero (as is the case for ERA)."

*6. In the manuscript, the "N" can be represented for several meanings, e.g., RO refractivity, the number of samples for statistic, and number of data sets, etc. It would be better to use different characters to avoid confusing.*

Thank you. We now use "*N*" for refractivity and "*n*" for the number of samples now throughout the manuscript.

*7. Page 4 line 15: There are three types of COSMIC data provided from CDAAC, i.e., re-processed, post-processed, and real-time data. In this manuscript, it uses the ERA-Interim data as the background for the 1DVAR retrieval, do the authors get the COSMIC data from re-processed data?*

We use only COSMIC re-processed data in this study. We added this information in Section 2.3 describing RO.

**Technical corrections:**
*1. The variables should be in the same form with italics, for example, page 4 line 17: "Specific humidity* q*" → please change to "Specific humidity q"; page 4 line 18: "water vapor pressure* e*" → "water vapor pressure e".*

Corrected

*2. Page 22 line 14 and line 16: three-cornered hat (THC) → three-cornered hat (TCH)*

Corrected to 3CH, which we now use throughout the paper.

*3. Pages 24 and 25: The descriptions for equations (13)-(15) are inconsistent, for example, the equation (13) is computed from Eqs. (3), (5) and (6), not (1), (2) and (3) that indicated in line 26.*

We have eliminated the derivation and associated discussion of the linearly dependent equations for estimating the error variances and so these equations no longer exist.

*4. Page 29 Fig. A4 figure caption: Mean of six...*
*of normalized "specific humidity" → should be "refractivity"*

Thank you. We have deleted this figure from the revised paper.

**Reviewer 2: Interactive comment on Atmos. Meas. Tech. Discuss., doi:10.5194/amt-2017-487, 2018**

*This manuscript applies the N-cornered hat technique to estimate errors in geophysical measurements: radio occultation, radiosondes, ERA-Interim reanalysis and weather forecast outputs at four locations in the tropics and subtropics. The N-cornered hat technique is closely related to the method of triple collocation, which has been widely applied to geophysical datasets in the literature (as the authors note). However, some subtle differences between the approaches are missed in this analysis, which may impact the results.*

*One important difference between triple collocation (TC) and the three-cornered hat (3CH) is the treatment of the underlying truth. TC treats the underlying truth as a random*

*variable (Stoffelen 1998), whereas 3CH does not. As a result, TC requires an additional assumption compared to 3CH: the errors must be uncorrelated with the underlying truth. Since the underlying truth is not considered to be a random variable in 3CH, the correlation between the errors and truth is always zero. So, should the underlying truth be treated as a random variable (as in TC) or as deterministic (as in 3CH)? I would argue that, for assessing the stability of clocks, the assumption of a deterministic underlying truth is quite reasonable. However, when considering atmospheric applications, as in this study, it is hard to justify. The atmosphere is a chaotic system with substantial internal variability. Differences between measurements can be due to measurement errors, but they can also be due to the internal variability of the chaotic system. In Figure 3a, for example, there are clear differences between the specific humidity profile estimated by GFS and RS on 13 January, 2007 at 00:23 UTC. Even if the measurement and modelling errors of both GFS and RS were both zero, we would expect there to be some difference between these two profiles because of the internal variability of the system, even accounting for some assimilation of observations into the GFS. Yet the 3CH implicitly attributes ALL differences between different measurement types to measurement errors in one or more measurement types. This seems misguided and is one of the reasons TC is typically applied to geophysical measurements rather than 3CH. Treating the underlying truth as deterministic rather than random leads the authors to neglect the impacts of possible covariance between the errors and the underlying truth, which is likely biasing the error estimates in this study.*

The 3CH method allows for a temporally varying "truth" and is appropriate for complex geophysical systems such as the atmosphere. For example, it has been successfully used by Valty et al. (2013) to estimate the geophysical load deformation computed from GRACE satellites, GPS vertical displacement measurements, and global general circulation (GCM) models. O'Carroll et al. (2008) estimated the errors associated with two radiometer and buoy measurements of sea-surface temperatures using exactly the same equations we used. They also provide an excellent discussion on "truth," and in particular conclude that the basic equations in the 3CH method (our Eqs. (7)-(9)) "continues to hold regardless of whatever our definition of the true value might be."

The differences associated with the "internal variability" of the atmosphere as illustrated in Fig. 3a are partly due to measurement errors and partly due to representativeness errors, as the reviewer notes. Both types of errors are included in the error estimates using the 3CH methods.

The main difference between the 3CH and TC method is that the TC method corrects for additive and multiplicative biases among the three data sets, as discussed by Stoffelen (1998), Vogelzang et al. (2011) and others. The TC method calibrates two of the data sets against the third, eliminating biases among the three data sets.

We have added a new section (A3) in Appendix A that compares the 3CH and TC method for a subset of our data sets. Also, Appendix A now includes more discussion of the differences between the two methods and the additive and multiplicative biases. The results in A3 of the Appendix show that the two methods give very similar results (please see next response).

*A second important difference is that TC accounts for multiplicative biases in the measurements in a way that 3CH does not. 3CH implies the following measurement model:*
*$i = T + e\_i$ . . . (1)*
*where T is the unknown true value, and e_i is the error in measurement i. In contrast, triple collocation starts with the measurement model:*
*$i = a\_i + b\_i*T + e\_i$ . . . (2)*
*where a_i and b_i are additive and multiplicative biases, respectively (see Gruber et al. (2016),*

*equation 1). The advantage of the measurement model in TC is that it is robust to multiplicative biases. If multiplicative biases exist but are not factored into the measurement model (as in 3CH), the multiplicative biases will lead to substantial correlations between the errors and the unknown truth. In turn, the neglect of these correlations will bias 3CH error estimates.*

*Given both of these problems with the analysis, it is not surprising that the estimated errors (for example, in Figure 10) are not internally consistent (as they should be, if either technique – 3CH or TC – were applied appropriately).*

We have applied the 3CH method appropriately. Our equations are correct and all assumptions are stated clearly. As mentioned above, the 3CH method has been applied successfully to other geophysical data sets. Biases in the measurements may be present, but tend to cancel as discussed in our discussion paper AMT-2018-75 "Evaluating two methods of estimating error variances from multiple data sets using an error model."

We do not consider the results shown in Fig. 10 to be inconsistent. Using different combinations of data sets to estimate the error variance profiles will lead to different estimates because of different (though small) error covariances in the different data sets and the limited sample size. This is also shown using the error model and idealized data sets in AMT-2018-75. We describe this in the summary and give more detail in Appendix A where the variation of three estimates is discussed: "If all the neglected error covariance terms were in fact identically zero and the sample size was very large (much larger than our sample size), all three estimates of the error variances would be the same. The fact that they give different solutions is because the neglected $COV_{err}$ terms are in reality not zero, and hence their neglect affects the three approximate equations in different ways to give three different solutions. The relatively small sample size $n$ also contributes to the differences in the three solutions, which are a measure of these effects."

Furthermore, as discussed above, we computed the estimated error variances for specific humidity using the TC method (including multiplicative and additive biases) and showed that the results are very similar to the 3CH method, confirming that the effect of biases is small in the 3CH method for these data sets. Please see two figures below for radio occultation (RO), radiosonde (RS), ERA, and GFS. We have added these results to Appendix A in the revised paper.

[Figure]

Fig. 1: Estimated RO and RS error variances for specific humidity at Minamidaitojima (Japan) using calibrated data as in the TC method (left) and the uncalibrated data as in the 3CH method (right). For the TC method, the RO, RS and GFS data sets are calibrated with respect to ERA as the reference data set. The following combinations of the 4 data sets are used: (ERA, RO, RS), (ERA, RO, GFS), and (ERA, GFS, RS).

[Figure]

Fig. 2: Estimated ERA and GFS specific error variances for ERA at Minamidaitojima (Japan) using the triple co-location (TC) method (left) and the three-cornered hat (THC) method (right). For the TC method, the RO, RS and GFS data sets are calibrated with respect to ERA. The following combinations of the 4 data sets are used: (ERA, RO, RS), (ERA, RO, GFS), and (ERA, GFS, RS).

*While 3CH and TC are similar in many respects, there are good reasons to use TC rather than 3CH when characterizing errors in geophysical data. Therefore, to address these concerns, I recommend the authors reframe their analysis using TC rather than 3CH. The differences between the techniques are relatively small, but significant, and warrant a substantial rewrite of the manuscript, in my view.*

It would be interesting to do a detailed comparison of the 3CH and TC methods. However, given the small differences of the error variances computed of the TC and 3CH methods (as depicted in Figs. 1 and 2 above), there is no reason to redo our entire study using a different method. The 3CH method is well established for geophysical systems (O'Carroll et al., 2008; Valty et al., 2013) and the realistic results shown in our paper speak for themselves. As discussed in the previous response, our error model in paper AMT-2018-75 showed that biases have a relatively small effect in the 3CH method, a result that is supported by the similarity in the results comparing the 3CH and TC methods (Figs. 1 and 2 above)

We have addressed this issue, as discussed above, by adding a new section in Appendix A that compares the TC and 3CH methods using a subset of our data, and shows that the two methods give very similar results.

*Specific comments Line 7, p 16: ". . .indicate that our estimates are reasonable and consistent with these studies." It seems unreasonable to be making this claim given the estimates in this study vary enormously. It is easy for an imprecise estimate to be consistent with previous studies, but this is not particularly informative.*

To our knowledge, this is this first study of its kind comparing multiple error variance estimates by using several different combinations of data sets at the same locations and time using the 3CH method. As far as we know, there are no previous studies of any kind that estimate vertical profiles of the errors associated with specific humidity, relative humidity, temperature and refractivity estimates over the entire range of the troposphere as done in our paper. In addition, as discussed in the response to Reviewer 1, the error characteristics of the models, radiosondes and RO are likely to vary with latitude, longitude, seasons and year (as the models change, radiosondes use different sensors, and even RO data and processing change). Thus there are no standards for comparison. However, as we show, the results we obtain are reasonable and consistent with other studies of individual observing systems and the consistency in the three estimates for each observations and the clear differences in the error estimates of the different data sets support the validity of the results.

This is a subtle point. The differences in the results using only RS, GFS, and ERA as pointed out above are due to slightly different sample sizes associated with the two RO retrievals in the mid and lower troposphere. There are two estimates of the error variances for RO because we estimate the error variance for RO using the Direct and 1D-VAR humidity retrievals of RO. This is discussed in the first paragraph of Section 5.1. For each RO retrieval, all four data sets (RS, RO, ERA, and GFS) are co-located. The data ensemble at each level is used only if data from all four sources are available on that level. The 1D-VAR RO data set extends to lower levels than the Direct RO data set because of the way the 1D-VAR is calculated. We now show this in a revised Figure 1 of the revised manuscript. Therefore, the number of data ensembles (including GFS, ERA and RS) available per level is slightly higher for all comparisons using the 1D-VAR retrieval compared to the number using Direct retrievals. These slightly different data sets give slightly different error variance estimates for the two (GFS,RS and ERA) ensembles.

*Page 15, line 26: "... mean of the six estimates of the error variances ...". But it looks like the light and dark results are separated. Is the mean only over three estimates each? How is the standard deviation with only three estimates taken? For small samples, it becomes important that the denominator in the expression for the sample variance is more correctly written as N-1 (N=3 here). This would also alter the shaded areas in the appendix.*

The reviewer is correct, this is the mean of the three estimates. We corrected the text on page 13 line 8 and the caption of Fig. 10.

Regarding the comment on how we computed the standard deviation, we used N, not N-1 in the original paper. We corrected this to use N-1 (two) in the revised paper (updated Figs. 10 and B1-B4). The results were not significantly different. We included the equation used to compute the STD in a footnote on page 13.

*Page 21, line 30: "... dry and wet water vapor biases ...". Needs reformulation.*

We reworded this to "...uses a radiosonde that is thought to have large water vapor biases...."

*Page 24, line 26: Shouldn't it be Eqs. (3), (5), and (6)? And the same for the following eqs.*

The reviewer is correct. However, we removed this entire section of the appendix (please see next comment).

*However, eqs. (10), (11), and (12), with (13), (14), and (15) inserted are not additional independent equations (as you also write). For example, (10) with (13) inserted is equivalent to (7) + (8) - (9). In general, you could get many more (infinitely many) equations if you don't care that they are dependent, namely A\*(7)+B\*(8)+C\*(9), where A, B, and C are any numbers, except those where A+B+C=0. Thus, there are not only six different ways. There are three independent ways and infinitely many if you also count dependent ones that can be formed by linear combinations of the first three. I think you need to make clear that there is nothing unique about the three additional equations that you choose. As it is, one could get the impression that they are in some way special (in line 5 you write "the full six equations" and in line 18 "the remaining three estimates"). Perhaps they are special if one can show that they give rise to the smallest possible standard deviation of the variance estimates. I don't know if that is the case. In the first three independent equations, there are 3/2 MS terms, and 3 COV terms involved. In the additional three equations there are 5/2 MS terms and 5 COV terms. The more terms there are, the larger the standard deviation of the variance estimates will become, at least potentially. Interesting stuff!*

We agree with this comment. And indeed the standard deviations of the estimated error variances using the three linearly dependent equations are greater than those from the three linearly dependent equations, as stated in line 6 page 29 of the original manuscript. However, we decided to eliminate the results from the three linearly dependent equations in the revised paper. We added the following paragraph in Appendix A:

"As noted by an anonymous reviewer, it is possible to derive infinitely many linearly dependent equations by combining Eqs. (A8)-(A10) in different ways using the form $M_1$ x Eq. (A8) + $M_2$ x Eq. (A9) + $M_3$ x Eq. (A10) where $M_1$ , $M_2$ and $M_3$ are any numbers except those for which $M_1 + M_2 + M_3 = 0$. We did not pursue this possibility in this paper, but instead used the three linearly independent equations only in our estimates of error variances."

*Page 25, line 15-16: I do not understand this part of the sentence: "...the set of observations in the pairs (RO,ERA), (RO,GFS), (GFS,ERA), (RO,RS), (RS,ERA) and (RS,GFS) are the same (they are in our case)..." What are the set of observations here?*

We clarified this by writing: "If all the neglected covariance terms were in fact identically zero and the sample size was very large (much larger than our sample size), all three estimates of the error variances would be the same."

*Technical corrections:*
*Page 2, line 3: model -> modeled*

Done

*Page 3, line 14: Missing "a" in front of global.*

Done

*Page 11, line 6: Acronym STD should be written out first time.*

Done

*Page 12, line 32: that -> than*

Done

*Page 15-16, lines 30,1-2: Should RS and RO be switched in the text here? RO is the one that oscillates between 0.1 and 0.3 (% squared). RS is fairly constant about 0.1.*

Yes. We corrected and clarified the description in the revised text.

*Page 21, line 17: Missing "to" in front of RO.*

Corrected

*Page 22, line 27: Vogelznang -> Vogelzang*

Corrected

*Page 25, line 18: "in are in"?*

A typo-we deleted the first "in".

**References used in our responses:**

Anlauf, H., Pingel, D., and Rhodin, A.: Assimilation of GPS radio occultation data at DWD, Atmos. Meas. Tech., 4, 1105–1113, https://doi.org/10.5194/amt-4-1105-2011, 2011.

Blackmore, T., O,Carroll, A., Saunders, R., and Aumann, H.H.: A comparison of sea surface temperature from the AATSR and AIRS instruments. Met. Office Forecasting Research Technical Report No. 495, 2007. (Available from nwp-publications@metoffice.gov.uk)

Burrows, C., Healy, S., and Culverwell, I.: Improving the bias characteristics of the ROPP refractivity

and bending angle operators, Atmos. Meas. Tech., 7, 3445–3458, https://doi.org/10.5194/amt-7-3445-2014, 2014.

Cucurull, L., Derber, J., and Purser, R.: A bending angle forward operator for global positioning system radio occultation measurements, J. Geophys. Res., 118, 14–28, https://doi.org/10.1029/2012JD017782, 2013.

Gilpin, S., Rieckh, T, and Anthes, R.: Reducing representativeness and sampling errors in radio occultation-radiosonde comparisons., Atmos. Meas. Tech., 11, 1-16, https://doi.org/10.5194/amt-11-1-2018, 2018a.

Gilpin, S., Anthes, R., and Sokolovskiy, S: Sensitivity of forward-modeled bending angles to vertical interpolation of refractivity for radio occultation data assimilation. To be submitted to Atmos. Meas. Tech., 2018b.

Healy, S. and Thépaut, J.-N.: Assimilation experiments with CHAMP GPS radio occultation measurements, Mon. Wea. Rev., 132, 605–623, 2006.

O'Carroll, A.G., Eyre, J.R., and Saunders, R.S: Three-way error analysis between AATSR, AMSR-E, and in situ sea surface temperature observations. J. Atmos. Oceanic Tech., 25,1197-1207, https://doi.org/10.1175/2007JTECHO542.1, 2008.

Valty, P., de Viron, O., Panet, I., Camp, M. V., and Legrand, J.: Assessing the precision in loading estimates by geodetic techniques in Southern Europe, Geophys. J. Int., 194, 1441–1454, https://doi.org/https://doi.org/10.1093/gji/ggt173, 2013.

**Estimating observation and model error variances using multiple data sets**

Richard Anthes[1] and Therese Rieckh[1,2]

[1]COSMIC Program Office, University Corporation for Atmospheric Research, Colorado, U.S.A.
[2]Wegener Center for Climate and Global Change, University of Graz, Graz, Austria

**Correspondence:** Richard Anthes (anthes@ucar.edu)

**Abstract.** In this paper we show how multiple data sets, including observations and models, can be combined using the "three cornered hatmethod"." (3CH) method to estimate vertical profiles of the errors of each system. Using data from 2007, we estimate the error variances of radio occultation, radiosondes, ERA-Interim, and GFS model data sets at four radiosonde locations in the tropics and subtropics. A key assumption is the neglect of error covariances among the different data sets, and we examine the consequences of this assumption on the resulting error estimates. Our results show that different combinations of the four data sets yield similar relative and specific humidity, temperature, and refractivity error variance profiles at the four stations, and these estimates are consistent with previous estimates where available. These results thus indicate that the correlations of the errors among all data sets are small and the 3CH method yields realistic error variance profiles. The estimated error variances of the ERA-Interim data set are smallest, a reasonable result considering the excellent model and data assimilation system and assimilation of high-quality observations. For the four locations studied, RO has smaller error variances than radiosondes, in agreement with previous studies.

**1 Introduction**

Estimating the error characteristics of any observational system or model is important for many reasons. Not only are these errors of scientific interest, they are important for data assimilation systems and numerical weather prediction. In many modern data assimilation schemes, observations of a given type are weighted proportionally to the inverse of their error variance (e.g. Desroziers and Ivanov, 2001).

Kuo et al. (2004) and Chen et al. (2011) used the difference between radio occultation (RO) observations of a variable X (e.g.refractivity) and short-range model forecasts of X to estimate the error of the RO observations, using the concept of apparent or perceived errors, defined by

$$X_{\mathrm{AE}} = X_{\mathrm{RO}} - X_{\mathrm{fcst}} \tag{1}$$

where $X_{\mathrm{AE}}$ is the apparent error of the RO observation and $X_{\mathrm{RO}}$ and $X_{\mathrm{fcst}}$ are the RO observations and model forecast values, respectively.

The error variance $\sigma_a^2$ of the apparent error is given by

$$\sigma_a^2 = \frac{1}{n} \sum \frac{X_{\text{AE}}^2}{N} X_{\text{AE}}^2 \qquad (2)$$

where  $n$ is the number of samples of observed and  modeled RO at the same location and time.

The relationship between the apparent error variance $\sigma_a^2$, the observational error variance $\sigma_o^2$, and the forecast error variance $\sigma_f^2$ is given by:

$$\sigma_a^2 = \sigma_o^2 + \sigma_f^2 - 2\text{COV}_{\text{err}}(X_{\text{RO}}, X_{\text{fcst}}) \qquad (3)$$

[revised manuscript text omitted]
}}(X) = \frac{1}{n} \sum (X - \underline{X_t}\text{True})^2 \underline{/N} = \frac{1}{n} \sum X_{\text{err}}^2 \tag{6}$$

[Figure]

**Figure 4.** Same as Fig. 3 except for normalized differences from ERA.

where  True is the true (but unknown) value of X and the summation is over  $n$ samples.

As shown in Appendix A, we can derive three different linearly independent equations for estimating the error variance of any data set, assuming that the error covariances among all the data sets are negligible compared to the differences in the observed mean square (MS) differences between the data sets.  The three complete (and exact) linearly independent solutions for estimating the error variance of RO are

$$2\text{VAR}(\text{RO} - \text{True}) = \text{VAR}_{\text{err}}(\text{RO}) = \text{MS}(\text{RO} - \text{ERA}) + \text{MS}(\text{RO} - \text{GFS}) - \text{MS}(\text{GFS} - \text{ERA})\underline{\text{MS}(\text{RO} - \text{ERA}) + \text{MS}(\text{RO} - \text{GFS}) - \text{MS}(\text{GFS} - }$$

$$+ \text{2 COV}(\text{RO}, \text{ERA}) + \text{COV}(\text{RO}, \text{GFS}) - \text{COV}(\text{GFS}, \text{ERA})2\left[\text{COV}_{\text{err}}(\text{RO}, \text{ERA}) + \text{COV}_{\text{err}}(\text{RO}, \text{GFS})\right.$$

$$(7)$$

$$2\text{VAR}(\text{RO} - \text{True}) = \text{VAR}_{\text{err}}(\text{RO}) = \text{MS}(\text{RO} - \text{ERA}) + \text{MS}(\text{RO} - \text{RS}) - \text{MS}(\text{RS} - \text{ERA})\underline{\text{MS}(\text{RO} - \text{ERA}) + \text{MS}(\text{RO} - \text{RS}) - \text{MS}(\text{RS} - \text{ERA}}$$

$$+ \text{2 COV}(\text{RO}, \text{ERA}) + \text{COV}(\text{RO}, \text{RS}) - \text{COV}(\text{RS}, \text{ERA})2\left[\text{COV}_{\text{err}}(\text{RO}, \text{ERA}) + \text{COV}_{\text{err}}(\text{RO}, \text{RS}) - \right.$$

$$(8)$$

$$2\mathrm{VAR(RO-True)} = \mathrm{VAR_{err}(RO)} = \mathrm{MS(RO-GFS) + MS(RO-RS) - MS(RS-GFS)}\;\mathrm{MS(RO-GFS) + MS(RO-RS) - MS(RS-GFS)]}$$
$$+\; 2\,\mathrm{COV(RO,GFS) + COV(RO,RS) - COV(RS,GFS)}\;2\left[\mathrm{COV_{err}(RO,GFS) + COV_{err}(RO,RS) - C}\right.$$

$$(9)$$

where RO (or ERA, GFS, RS) corresponds to the value of X as estimated by RO (or ERA, GFS, RS) , True corresponds to the true (but unknown) value of X and MS denotes the mean square difference between the values from two data sets (e.g.

RO – ERA).

We use Eq. (7)–(9) to provide three independent estimates of VAR(RO – True$_{err}$(RO) by neglecting the COV$_{err}$ terms in each equation. The assumption that the error covariances are small compared to the difference in variances between the data sets is similar to the assumption used in the apparent error method that the errors of the observations and model forecasts are uncorrelated. Of course in In general the COV$_{err}$ terms are not zero; thus we will examine the validity of this assumption by checking whether the various estimates of the error variances from the three equations are consistent with each other and reasonable compared to other independent studies that estimate error variances in other ways. In a related paper (Rieckh and Anthes, 2018) we examine the effect of various degrees of error correlations between two of the three data sets using an error model.

[revised manuscript text omitted]

10    (use of GFS temperature in Eq. (5)) shows a steady increase of error variance with height, from about $100\,\%^2$ (STD $\sim$10 %) at $950\,\mathrm{hPa}$ to $800\,\%^2$ (STD $\sim$28 %) at $500\,\mathrm{hPa}$ and $2000\,\%^2$ (STD $\sim$45 %) at $300\,\mathrm{hPa}$. This is expected since the refractivity contains little information on water vapor above about $400\,\mathrm{hPa}$ and we are using an independent estimate of temperature, with no constraints on the water vapor retrieval. The $q$ error variance profile for RO using the 1D-VAR method is similar to that of the direct method below $500\,\mathrm{hPa}$, but reaches a maximum at about $500\,\mathrm{hPa}$ of about $500\,\%^2$ (STD $\sim$22 %) and then decreases

[revised manuscript text omitted]

---

## Author Response (AR3)

**Response to Associate editor comments on AMT 2017-487 "Estimating observation and model error variances using multiple data sets"**

**Associate Editor comments in italics are given below, followed by our responses:**

*All reviewer and associate editor comments have been well considered, putting this manuscript well in context with existing scientific literature.*

*A last minor note is that according to the Oxford dictionary, collocation has the meaning of "The action of placing things side by side or in position". Please consider to use it rather than co-location.*

We found the following definitions for collocate and co-locate
([www.merriam-webster.com/dictionary/](www.merriam-webster.com/dictionary/)):

**Collocate:**   transitive verb: to set or arrange in a place or position; especially: to set side by side
                  intransitive verb: to occur in conjunction with something

**Colocate or co-locate:** transitive + intransitive: to locate (two or more things) together or be located together: such as:
          - to cause (two or more things) to be in the same place or close together
          - to place (two or more units) close together so as to share common facilities
          - to place (computer servers) together in a secure dedicated storage facility
          -  be in the same location

A number of articles discuss the use of "collocate" versus "co-locate" and we found that "co-locate" is more appropriate in the context of our manuscript.

We therefore kept the word "co-locate" in the manuscript but changed it to "collocate" when referring to the "Triple collocation method" to have consistent spelling with Stoffelen (1998).